# `einspace`: Searching for Neural Architectures from Fundamental Operations

**Linus Ericsson**[1]* **Miguel Espinosa**[1] **Chenhongyi Yang**[1] **Antreas Antoniou**[2]
**Amos Storkey**[2] **Shay B. Cohen**[2] **Steven McDonagh**[1] **Elliot J. Crowley**[1]

[1] School of Engineering [2] School of Informatics
University of Edinburgh

Project page: `https://linusericsson.github.io/einspace`
Code: `https://github.com/linusericsson/einspace`

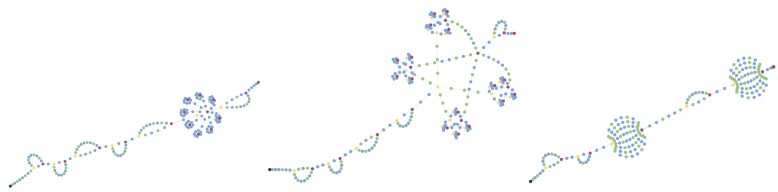

## Abstract

Neural architecture search (NAS) finds high performing networks for a given task. Yet the results of NAS are fairly prosaic; they did not e.g. create a shift from convolutional structures to transformers. This is not least because the search spaces in NAS often aren't diverse enough to include such transformations *a priori*. Instead, for NAS to provide greater potential for fundamental design shifts, we need a novel expressive search space design which is built from more fundamental operations. To this end, we introduce `einspace`, a search space based on a parameterised probabilistic context-free grammar. Our space is versatile, supporting architectures of various sizes and complexities, while also containing diverse network operations which allow it to model convolutions, attention components and more. It contains many existing competitive architectures, and provides flexibility for discovering new ones. Using this search space, we perform experiments to find novel architectures as well as improvements on existing ones on the diverse Unseen NAS datasets. We show that competitive architectures can be obtained by searching from scratch, and we consistently find large improvements when initialising the search with strong baselines. We believe that this work is an important advancement towards a transformative NAS paradigm where search space expressivity and strategic search initialisation play key roles.

## 1 Introduction

The goal of neural architecture search (NAS) [14, 42] is to automatically find a network architecture for a given task, removing the need for expensive human expertise. NAS uses (i) a defined search space of all possible architectures that can be chosen, and (ii) a search algorithm e.g. [68, 58, 40] to navigate through the space, selecting the most suitable architecture with respect to search objectives. Despite significant research investment in NAS, with over 1000 papers released since 2020 [59], manually designed architectures still dominate the landscape. If someone looks through recent deep learning papers, they will most likely come across a (manually designed) transformer [56], or perhaps a (manually designed) MLP-Mixer [53], or even a (manually designed) ResNet [19]. Why isn't NAS being used instead?

---

*linus.ericsson@ed.ac.uk

38th Conference on Neural Information Processing Systems (NeurIPS 2024).

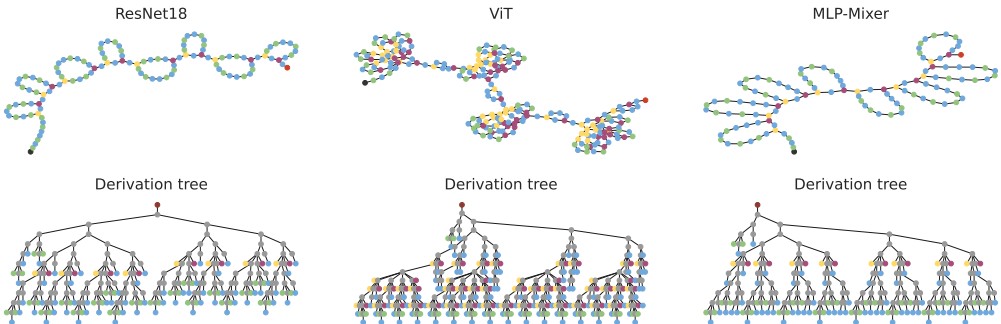

Figure 1: Three state-of-the-art architectures and their associated derivation trees within `einspace`. Top row shows the architectures where the black node is the input tensor and the red is the output. Bottom row shows derivation trees where the top node represents the starting symbol, the grey internal nodes the non-terminals and the leaf nodes the terminal operations. See Section 3.1 for details on other node colouring. Best viewed with digital zoom.

Part of the problem is that most NAS search spaces are not expressive enough, relying heavily on high-level operations and rigid structures. For example in the DARTS [30] search space, each architecture consists of repeating cells; each cell is a directed acyclic graph where nodes are hidden states, and edges are operations drawn from a fixed set of mostly convolutions. This encodes a very specific prior—*architectures contain convolutions with multi-scale, filter-like structures* [50]—making it impossible to discover anything beyond ConvNet characteristics. Indeed, random search [63, 27] is often a strong baseline in NAS; networks sampled from unexpressive spaces behave very similarly [57] which makes it hard to justify an (often expensive) search.

One solution is to take the *ex nihilo* approach to search space construction. In AutoML-Zero [41] the authors create a very expressive search space that composes basic mathematical operations without any additional priors. However, searching through this space is far too expensive for mainstream use, requiring several thousand CPUs across several days to (re)discover simple operations like linear layers and ReLUs. Recent interest in hierarchical search spaces [47] has enabled the study of search across differing architectural granularities which naturally allows for greater flexibility. However, attempts so far have been limited to single architecture families like ConvNets [47] or transformers [67]. The hybrid search spaces that do exist have limited options both on the operation-level and macro structure [26, 61].

For NAS to be widely used we need the best of both worlds: a search space that is both highly expressive, and in which we can straightforwardly use existing tried-and-tested architectures as powerful priors for search. To this end, we propose `einspace`: a neural architecture search space based on a parameterised probabilistic context-free grammar (CFG). It is *highly expressive*, able to represent diverse network widths and depths as well as macro and micro structures. With its expressivity, the space contains disparate state-of-the-art architectures such as ResNets [19], transformers [56, 13] and the MLP-Mixer [53], as shown in Figure 1. Other notable architectures contained in `einspace` are DenseNet [23], WideResNet (WRN) [64], ResMLP [54] and the Vision Permutator [21].

We realise our proposed search space through the creation of function-mapping groups that define a broad class of fundamental network operations and further describe how such elements can be composed into full architectures under the natural recursive capabilities of our CFG. To guarantee the validity of all architectures generated within the expressive space, we first extend our base CFG with parameters that ensure diverse components can be combined into complex structures. Next, we balance the contention between search space *flexibility* and search space *complexity* by introducing mild constraints on our search space via branching and symmetry-based priors. Finally, we integrate probabilities into our production rules to further control the complexity of architectures sampled from our space.

To demonstrate the effectiveness of `einspace`, we perform experiments on the Unseen NAS datasets [16]—eight diverse classification tasks including vision, language, audio, and chess problems—using simple random and evolutionary search strategies. We find that in such an expressive search space, the choice of search strategy is important and random search underperforms. When using the powerful priors of human-designed architectures to initialise the search, we consistently find

both large performance gains and significant architectural changes. Code to reproduce our experiments is available at `https://github.com/linusericsson/einspace`.

Using only simple search strategies, we can still identify competitive architectures, indicating that further refining these strategies in `einspace` could lead to significant advancements. We hope that this novel perspective on search spaces—focusing on expressiveness and incorporating the priors of existing state-of-the-art architectures—has the potential to drive NAS research towards a new paradigm.

## 2 Background

**Neural architecture search**
The search space used in NAS has a significant impact on results [63, 65]. This has facilitated the need to investigate search space design alongside the actual search algorithms [37]. Early macro design spaces [24, 68] made use of naive building blocks while accounting for skip connections and branching layers. Further design strategies have looked at chain-structured [3, 4, 6, 43], cell-based [69, 66, 30, 15] and hierarchical approaches. Hierarchical search spaces have been shown to be expressive and effective in reducing search complexity and methods include factorised approaches [52], $n$-level hierarchical assembly [28, 29, 48], parameterisation of hierarchical random graph generators [45] and topological evolutionary strategies [35]. Additional work on search spaces have proposed new candidate operations and module designs such as hand-crafted multi-branch cells [51], tree-structures [3], shuffle operations [32], dynamic modules [22], activation functions [38] and evolutionary operators [10]. In AutoML-Zero [41], the authors try to remove human bias from search space construction by defining a space of basic mathematical operations as building blocks.

The pioneering work of [47] constructs search spaces using CFGs. We take this direction further and construct `einspace` as a probabilistic CFG allowing for unbounded derivations, balanced by careful tuning of the branching rate. We aim to strike a balance between the level of complexity in the search space and incorporating components from diverse state-of-the-art architectures. Crucially, our space enables flexibility in both macro structure and at the individual operation level. While previous search spaces can be instantiated for specific architecture classes[30, 12], our single space incorporates multiple classes in one, ConvNets, transformers and MLP-only architectures. Such hybrid spaces have been explored before [26], but they have been limited in their flexibility, offering only direct choices between convolution and attention operations and disallowing the construction of novel components.

Prominent search strategies employed for NAS include Bayesian optimisation [34, 58], reinforcement learning [66, 68, 69] and genetic algorithms [5, 39, 40]. A popular thread of work, towards improving computational efficiency via amortising training cost, involves the sharing of weights between different architectures via a supernet [1, 3, 9, 18, 30, 31]. Efficiency has been further improved by sampling only a subset of supernet channels [60], thus reducing both space exploration redundancies and memory consumption. Alternative routes to mitigating space requirements have considered both architecture and operation-choice pruning [7, 15]. We however highlight that random search often proves to be a very strong baseline [27, 63]; a consequence of searching within narrow spaces. This is commonly the case for highly engineered search spaces that contain a high fraction of strong architectures [59]. Contrasting this, in our `einspace` we observe that random search across many tasks performs poorly, underpinning the value of a good search strategy for large, diverse search spaces [2, 41].

**Context-free grammars**
A context-free grammar (CFG; [20]) is a tuple $(N, \Sigma, R, S)$, where $N$ is a finite set of *non-terminal* symbols, $\Sigma$ is a finite set of *terminal* symbols, $R$ is the set of *production rules*—where each rule maps a non-terminal $A \in N$ to a string of non-terminal or terminals $A \to (N \cup \Sigma)^+$—and $S$ is the *starting symbol*. A CFG describes a context-free language, containing all the strings that the CFG can generate. By recursively selecting a production, starting with the rules containing the starting symbol, we can generate strings within the grammar. CFGs can be *parameterised*: each non-terminal, in each rule in $R$, is annotated with parameters $p_1, \ldots, p_n$ that influence the production. These parameters can condition production, based on an external state or contextual information, thus extending the power of the grammar.

A *probabilistic* context-free grammar (PCFG) associates each production rule with a probability [33]. These define the likelihood of selecting a particular rule given a parent non-terminal. The assigned probabilities allow for stochastic string generation.

# 3   `einspace`: A Search Space of Fundamental Operations

Our neural architecture search space, `einspace`[2], is introduced here. Based on a parameterised PCFG, it provides an expressive space containing many state-of-the-art neural architectures. We first describe the groups of operations we include in the space, then how macro structures are represented. We then present the CFG that defines the search space and its parameterised and probabilistic extensions.

As a running example we will be constructing a simple convolutional block with a skip connection within `einspace`, explaining at each stage how it relates to the architecture. The block will consist of a convolution, a normalisation and an activation, wrapped inside a skip connection.

## 3.1   Fundamental Operations

Each fundamental operation within `einspace` takes as input a tensor, either passed as input to the whole network or an intermediate tensor from a previous operation, and operates on it further. An operation can be thought of as a *layer* in a processing pipeline that defines the overall network. The fundamental operations can be separated into four distinct groups of functions that define their role in a network architecture. The terms *one-to-one*, *one-to-many* and *many-to-one* below refer to the number of input and output tensors of the functions within that group. For full details of the operations, see Appendix A.2

**Branching**. *One-to-many* functions that direct the flow of information through the network by cloning or splitting tensors. Examples include the branching within self-attention modules into queries, keys and values. In our visualisations, these are coloured **yellow**.

**Aggregation**. *Many-to-one* functions that merge the information from multiple tensors into one. Examples include matrix multiplication, summation and concatenation. In our visualisations, these are coloured **purple**.

**Routing**. *One-to-one* functions that change the shape or the order of the content in a tensor without altering its information. Examples include axis permutations as well as the `im2col` and `col2im` operations. In our visualisations, these are coloured **green**.

**Computation**. *One-to-one* functions that alter the information of the tensor, either by parameterised operations, normalisation or non-linearities. Examples include linear layers, batch norm and activations like ReLU and softmax. In visualisations, these are coloured **blue**.

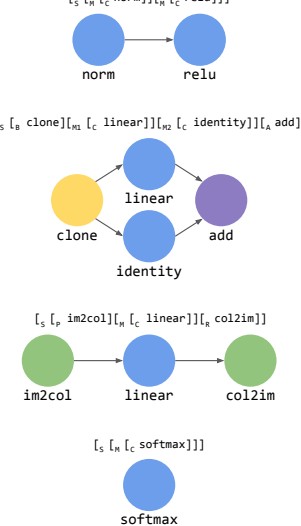

In our example, the skip connection will be handled by a combination of branching and aggregation functions, the convolution is decomposed into the routing functions `im2col` and `col2im`, with a `linear` layer from the computation group between them. The normalisation and activation come from the computation group. In the next subsection, we discuss the larger structures of the architecture.

## 3.2   Macro Structure

The groups of functions above describe the fundamental operations that make up an architecture. We now describe how these functions are composed in different ways to form larger components.

A *module* is defined as a composition of functions from above that takes one input tensor and produces one output tensor, with potential branching inside. A module may contain multiple *computation* and *routing* operations, but each *branching* must be paired with a subsequent *aggregation* operation. Thus, the whole network can be seen as a module that takes a single tensor as input and outputs a single prediction. A network module may itself contain multiple modules, directly pertaining to the hierarchical phrase nature of CFG structures. We divide modules into four types, visualised in Figure 2.

Figure 2: Visualisation of example modules with their CFG derivations in bracket notation. From top to bottom; sequential, branching, routing and computation modules.

---

[2]The name is inspired by the generality of *Einstein summation* and the related Python library `einops` [44] as many of our operations can be implemented by it.

**Sequential module**. A pair of modules and/or functions that are applied to the input tensor sequentially. Using our grammar, defined in Section 3.3, this can be produced using the rule (M→MM), or equivalently from the starting symbol S. This also applies to the rules below.

**Branching module**. A branching function first splits the input into multiple branches. Each branch is processed by some inner set of modules and/or functions. The outputs of all branches are subsequently merged in an aggregation function. In the grammar below this can be produced by the rule (M→B M A).

**Routing module**. A routing function is applied, followed by a module and/or function. A final routing function then processes the output tensor. In the grammar below this is produced by the rule (M→P M R). For more details on the role of the routing module, see Appendix A.3.

**Computation module**. This module only contains a single function, selected from the one-to-one computation functions described above. While this module is trivial, we will see later how its inclusion is helpful when designing our CFG and its probabilistic extension. In the grammar below this is produced by the rule (M→C).

To construct our example, we will be using all four modules. The branching module combines the `clone` and `add` functions from before to create a 2-branch structure. One branch is a simple skip connection by using the `identity` function inside a computation module. The other branch is the more complex sequence. The convolutional layer is created by combining `im2col`, `linear` and `col2im` in a routing module. The norm and activation are each wrapped in a computation module and these are all composed in sequential modules. Figure 2 shows similar module instantiations in action.

### 3.3 Search Space as a Context-Free Grammar

The following CFG defines our `einspace`, where uppercase symbols represent non-terminals and lowercase represent terminals. The colours refer to the function groups.

```
S  →  M  M  |  B  M  A  |  P  M  R,
M  →  M  M  |  B  M  A  |  P  M  R  |  C,
B  →  clone  |  group,
A  →  matmul  |  add  |  concat,
P  →  identity  |  im2col  |  permute,
R  →  identity  |  col2im  |  permute,
C  →  identity  |  linear  |  norm  |  relu  |  softmax  |  pos-enc.
```

Our networks are all constructed according to the high-level blueprint: `backbone`→`head` where `head` is a predefined module that takes an output feature from the backbone and processes it into a prediction (see Appendix B for more details). The `backbone` is thus the section of the network that is generated by the above CFG. When searching for architectures we search across different backbones.

Completing our running example, we present the full derivation of the architecture in the CFG in Figure 3.

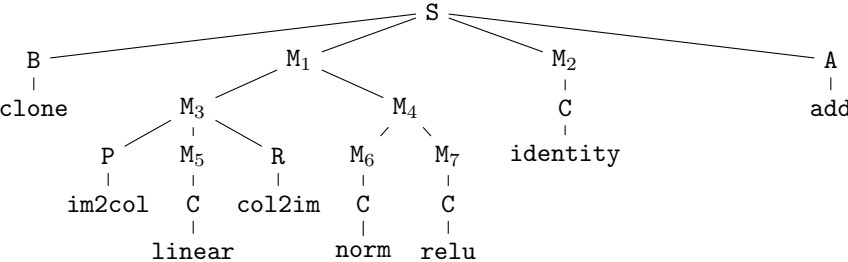

Figure 3: Example derivation tree of a traditional convolutional block with a skip connection.

### 3.4 Prior Choices

When designing a search space, we must balance the need for flexibility—which allows more valid architectures to be included—and constraints – which reduce the size of the search space. We can view constraints as imposing priors on which architectures we believe are worth including. As discussed, many previous frameworks are too restrictive; therefore, we aim to impose minimal priors, listed below.

**Convolutional prior**. We design our routing module to enable convolutions to be easily constructed, while also allowing components like patch embeddings and transpose operations. We thus enforce that a routing function is followed by another routing function later in the module. Moreover, `im2col` only appears in the production rule of the first routing function (`P`) and `col2im` in the last (`R`). As shown in Figure 2, to construct a convolution, we start from the rule (`M`→`P M R`) and derive the following (`P`→`im2col`), (`M`→`C`→`linear`) and (`R`→`col2im`).

**Branching prior**. We also impose a prior on the types of branching that can occur in a network. The branching functions `clone` and `group` can each have a branching factor of 2, 4 or 8. For a factor of 2, we allow each inner function to be unique, processing the two branches in potentially different ways. For branching factors of 4 or 8, the inner function `M` is repeated as is, processing all branches identically (though all inner functions are always initialised with separate parameters). Symbolically, given a branching factor of 2 we have (`B`$M_1$ $M_2$ `A`) but with a branching factor of 4 we have (`B`$M_1$ $M_1$ $M_1$ $M_1$ `A`). Examples of components instantiated by a branching factor of 2 include skip connections, and for 4, or 8, multi-head attention.

### 3.5 Feature Mode

Different neural architectures operate on different feature shapes. ConvNets maintain 3D features throughout most of the network while transformers have 2D features. To enable such different types of computations in the same network, we introduce the concept of a *mode* [3] that affects the shape of our features and which operations are available at that point in the network. Before and after each module, we fix the feature tensor to be of one of two specific shapes, depending on which mode we are in.

**Im mode**. Maintains a 3D tensor of shape (`C, H, W`), where `C` is the number of channels, `H` is the height and `W` is the width. Most convolutional architectures operate in this mode.

**Col mode**. Maintains a 2D tensor of shape (`S, D`), where `S` is the sequence length and `D` is the token dimensionality. This is the mode in which most transformer architectures operate.

The mode is changed by the routing functions `im2col` and `col2im`. Most image datasets will provide inputs in the `Im` mode, while most tasks that use a language modality will provide it in `Col` mode.

Our example architecture maintains the `Im` mode at almost all stages, apart from inside the routing modules where the `im2col` function briefly puts us in the `Col` mode before `col2im` brings us back.

### 3.6 Parameterising the Grammar

Due to the relatively weak priors we impose on the search space, sampling a new architecture naively will often lead to invalid networks. For example, the shape of the output tensor of one operation may not match the expected input shape of the next. Alternatively, the branching factor of a branching function may not match the branching factor of its corresponding aggregation function.

We therefore extend the grammar with parameters. Each rule $r$ now has an associated set of parameters $(s,m,b)$ that defines in which situations this rule can occur. When we sample an architecture from the grammar, we start by assigning parameter values based on the expected input to the architecture. For example, they might be the input tensor shape, feature mode and branching factor:

$$(s=[3,224,224],m=\texttt{Im},b=1). \tag{1}$$

Given this, we can continuously infer the current parameters during each stage of sampling by knowing how each operation changes them. When we expand a production rule, we must choose a rule which has matching parameters. If at some point, the sampling algorithm has no available valid options, it

---

[3]Note that this is similar but not the same as the *mode* of a general tensor, which determines the number of dimensions of that tensor. We use the term mode to refer to the state that a particular part of the architecture is in.

will backtrack and change the latest decision until a full valid architecture is found. Hence, we ensure that we can sample architectures without risk of obtaining invalid ones.

As an example of this, the CFG rule for P was previously

$$\texttt{P} \quad \rightarrow \quad \texttt{identity} \quad | \quad \texttt{im2col} \quad | \quad \texttt{permute.} \tag{2}$$

Enhanced with parameters, this now becomes two rules

$$\texttt{P}(m\!=\!\texttt{Im}) \quad \rightarrow \quad \texttt{identity} \quad | \quad \texttt{im2col} \quad | \quad \texttt{permute}, \tag{3}$$
$$\texttt{P}(m\!=\!\texttt{Col}) \quad \rightarrow \quad \texttt{identity} \quad | \quad \texttt{permute.} \tag{4}$$

This signifies that an `im2col` operation is not available in the `Col` mode. Similarly, the available aggregation options depend on the branching factor of the current branching module

$$\texttt{A}(b\!=\!2) \quad \rightarrow \quad \texttt{matmul} \quad | \quad \texttt{add} \quad | \quad \texttt{concat}, \tag{5}$$
$$\texttt{A}(b\!=\!4) \quad \rightarrow \quad \texttt{add} \quad | \quad \texttt{concat}, \tag{6}$$
$$\texttt{A}(b\!=\!8) \quad \rightarrow \quad \texttt{add} \quad | \quad \texttt{concat.} \tag{7}$$

### 3.7 Balancing Architecture Complexity

When sampling an architecture, we construct a decision tree where non-leaf nodes represent decision points and leaf nodes represent architecture operations. In each iteration, we either select a non-terminal module to expand the architecture and continue sampling, or choose a terminal function to conclude the search at that depth. Continuously selecting modules results in a deeper, more complex network, whereas selecting functions leads to a shallower, simpler network. We can balance this complexity by assigning probabilities to our production rules, thereby making a PCFG. Recall our CFG rule

$$(\texttt{M} \quad \rightarrow \quad \texttt{M M} \quad | \quad \texttt{B M A} \quad | \quad \texttt{P M R} \quad | \quad \texttt{C}). \tag{8}$$

If we choose one of the first three options we are delving deeper in the search tree since there is yet another M to be expanded, but if we choose ( M → C ), the *computation-module*, then we will reach a terminal function. Thus, to balance the depth of our traversal and therefore expected architecture complexities, we can set probabilities for each of these rules:

$$p(\texttt{M}\!\rightarrow\!\texttt{M M}\,|\,\texttt{M}), \quad p(\texttt{M}\!\rightarrow\!\texttt{B M A}\,|\,\texttt{M}), \quad p(\texttt{M}\!\rightarrow\!\texttt{P M R}\,|\,\texttt{M}), \quad p(\texttt{M}\!\rightarrow\!\texttt{C}\,|\,\texttt{M}). \tag{9}$$

The value of $p(\texttt{M}\!\rightarrow\!\texttt{C}\,|\,\texttt{M})$ is especially important as it can be interpreted as the probability that we will stop extending the search tree at the current location.

We could set these probabilities to match what we wish the expected depth of architectures to be (for empirical results on the architecture complexity, see Table 10 in the Appendix). However, we can actually ensure that the CFG avoids generating infinitely long architecture strings by setting the probabilities such that the branching rate of the CFG is less than one [8]. For details of how, see Appendix A.4. So, as shown in Figure 4, we set the computation module probability to $p(\texttt{M}\!\rightarrow\!\texttt{C}\,|\,\texttt{M})\!=\!0.32$ and the probabilities of the other modules to $\frac{1-0.32}{3}$. For simplicity, all other rule probabilities are uniform.

For a thorough example of how sampling is performed in `einspace`, please see Appendix A.1.

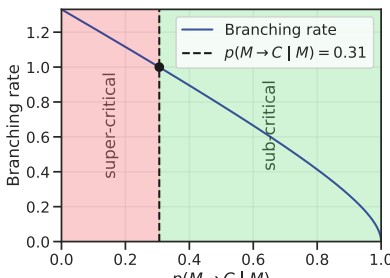

Figure 4: To ensure our CFG is consistent and does not generate infinite architectures, we make sure the branching rate is in the sub-critical region by setting $p(\texttt{M}\!\rightarrow\!\texttt{C}\,|\,\texttt{M})\!>\!0.31$.

## 4 Experiments

### 4.1 Experimental Details

**Datasets**
As our search space strives for expressivity and diverse architectures, we adopt a diverse benchmark suite from the recent paper on Unseen NAS [16], containing datasets at different difficulties across

vision, language, audio and further modalities. We run individual searches on these datasets, that are each split into train, validation and test sets. See Appendix B.2 for the detailed dataset descriptions.

While Unseen NAS forms the basis of this section, we run additional experiments on the diverse NAS-Bench360 benchmark [55] in Appendix C.1, where we beat competing NAS methods on CIFAR100, FSD50K and Darcy Flow, and to the best of our knowledge set a new state-of-the-art on NinaPro.

**Search strategy**

We explore three search strategies within `einspace`: random sampling, random search, and regularised evolution (RE). *Random sampling* estimates the average expected test performance from $K$ random architecture samples. *Random search* samples $K$ architectures and selects the best performer on a validation set. In *regularised evolution*, we start by constructing an initial population of 100 individuals, which are either randomly sampled from the search space or seeded with existing architectures. For $(K - 100)$ iterations, the algorithm then randomly samples 10 individuals and selects the one with the highest fitness as the parent. This parent is mutated to produce a new child. This child is evaluated and enters the population while the oldest individual in the population is removed, following a first-in-first-out queue structure. An architecture is mutated in three straightforward steps:

1. *Sample a Node*: Uniformly at random sample a node in the architecture derivation tree.

2. *Resample the Subtree*: Replace the subtree rooted at the sampled node by regenerating it based on the grammar rules. This step allows the exploration of new configurations, potentially altering a whole subtree if a non-leaf node is chosen.

3. *Validate Architecture*: Check if the new architecture can produce a valid output in the forward pass, given an input of the expected shape, and that it fits within resource constraints, e.g. GPU memory. If it does, accept it; otherwise, discard and retry the mutation.

Note that these are very simple search strategies, and that there is huge potential to design more intelligent approaches, e.g. including crossover operations in the evolutionary search, using hierarchical Bayesian optimisation [47] or directly learning the probabilities of the CFG [11]. In this work, we focus on the properties of our search space and investigate whether simple search strategies are able to find good architectures, and leave investigations on more complex search strategies for future work.

**Baselines**

We compare these search strategies to PC-DARTS [60], DrNAS [7] and Bonsai-Net [15] with results transcribed from [16]. We also compare to the performance of a trained ResNet18 (RN18). More details on the baselines, training recipes and network instantiations can be found in Appendix B.1

## 4.2 Random Sampling and Search

In previous NAS search spaces e.g. [30, 62, 12], complex search methods often perform very similarly to random search [27, 63]. Indeed, we can see this in Table 1 comparing the PC-DARTS strategy to DARTS random search.

However for `einspace`, this is not the case for most datasets. Random sampling improves on pure random guessing (not shown), but is far from the baseline performance of a ResNet18. The random search baseline is also far behind, but intriguingly outperforms baseline NAS approaches on Chesseract.

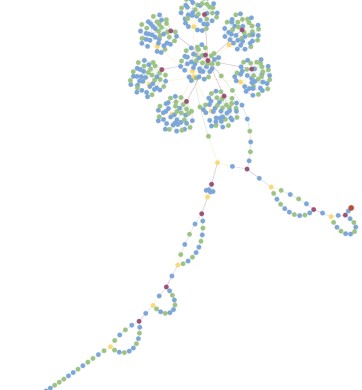

Figure 5: The top RE(Mix) architecture on AddNIST, found in `einspace`.

## 4.3 Evolutionary Search from Scratch

We now turn to a more sophisticated search strategy. We perform regularised evolution in `einspace` for 1000 iterations across all datasets, initialising the population with 100 random samples. In Table 1 the results are shown in the column named RE(Scratch). The performance of this strategy is significantly higher than random search on several datasets, indicating that the search strategy is more important in an expressive search space like `einspace` compared to DARTS. Compared to the top performing NAS methods, however, it is significantly behind on some datasets.

Table 1: Accuracies resulting from the combination of `einspace` with the simple search strategies of random sampling, random search, and regularised evolution (RE). See text for further detail. We evaluate performance across multiple datasets and modalities from Unseen NAS [16]. Results transcribed from [16] are denoted *, where DARTS [30] and Bonsai [15] search spaces are employed. The expressiveness of `einspace` enables performance that remains competitive with significantly more elaborate search strategies, as well as outperforming the CFG-based space hNASBench201 [46] when using evolutionary search in both spaces. **Best** and second best performance per dataset.

| | Baselines | | | | Regularised evolution (RE) | | | | Rand. Search | | | Rand. Sampl. |
| | | | | | hNB201 | einspace | | | | | | |
| Dataset | RN18 | PC-DARTS* | Dr NAS* | Bonsai-Net* | RE (Scratch) | RE (RN18) | RE (Mix) | RE (Scratch) | DARTS* | Bonsai* | ein space | ein space |
|---|---|---|---|---|---|---|---|---|---|---|---|---|
| AddNIST | 93.36 | 96.60 | 97.06 | **97.91** | 93.82 | 97.54 | 97.72 | 83.87 | 97.07 | 34.17 | 67.00 | 10.13 |
| Language | 92.16 | 90.12 | 88.55 | 87.65 | 92.43 | 96.84 | **97.92** | 88.12 | 90.12 | 76.83 | 87.01 | 35.26 |
| MultNIST | 91.36 | 96.68 | **98.10** | 97.17 | 93.44 | 96.37 | 92.25 | 93.72 | 96.55 | 39.76 | 66.09 | 18.87 |
| CIFARTile | 47.13 | **92.28** | 81.08 | 91.47 | 58.31 | 60.65 | 62.76 | 30.89 | 90.74 | 24.76 | 30.90 | 25.25 |
| Gutenberg | 43.32 | 49.12 | 46.62 | 48.57 | 43.70 | **54.02** | 50.16 | 36.70 | 47.72 | 29.00 | 39.58 | 19.69 |
| Isabella | 63.65 | 65.77 | 64.53 | 64.08 | 59.79 | 64.30 | 62.72 | 56.33 | **66.35** | 58.53 | 56.90 | 32.24 |
| GeoClassing | 90.08 | 94.61 | **96.03** | 95.66 | 92.33 | 95.31 | 95.13 | 60.43 | 95.54 | 63.56 | 69.13 | 24.35 |
| Chesseract | 59.35 | 57.20 | 58.24 | 60.76 | 63.92 | 60.31 | 61.86 | 59.50 | 59.16 | **68.83** | 61.46 | 44.83 |
| Average acc. ↑ | 72.55 | 80.30 | 78.78 | **80.41** | 74.72 | 78.17 | 77.56 | 63.70 | **80.41** | 49.43 | 59.76 | 26.33 |
| Average rank ↓ | 7.38 | 4.69 | 4.62 | **3.75** | 6.00 | 3.88 | 4.12 | 9.00 | 4.31 | 9.50 | 8.88 | 11.88 |

## 4.4 Evolutionary Search from Existing SOTA Architectures

To fully utilise the powerful priors of existing human-designed structures, we now invoke search where the initial population of our evolutionary search is seeded with a collection of existing state-of-the-art architectures.

We first seed the entire population with the ResNet18 architecture. The search applies mutations to these networks for 500 iterations. In Table 1, these results can be found in the RE(RN18) column.

To further highlight the expressivity of `einspace`, we perform experiments searching from an initial population seeded with a mix of ResNet18, WRN16-4, ViT and MLP-Mixer architectures. To our knowledge, no other NAS space is able to represent such a diverse set of architectures in a single space. These results are shown in the RE(Mix) column.

Overall, we find that on **every single task**, we can find an improved version of the initial architecture using RE(RN18) and on all but one using RE(Mix). Moreover, in some cases we can beat the existing state-of-the-art, especially on tasks further from the traditional computer vision setting. In particular,

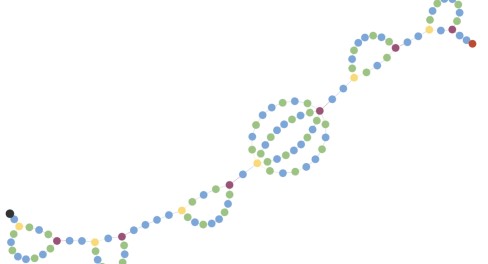

Figure 6: The best model on the Language dataset, found by RE(Mix) in `einspace`.

where previous NAS methods fail—i.e. the Language dataset—the architecture in Figure 6 has a direct improvement over the ResNet18 by 5.76%. See also the architecture in Figure 5 and the collection in Figure 8 in the Appendix for the breadth of structures that are found in `einspace`.

We further compare `einspace` to the previous, CFG-based, hNASBench201 from Schrodi et al. [46]. This allows for an initial study on the effects of our search space design choices and, in particular, the increased expressiveness compared to hNASBench201. These results show how `einspace` compares favourably to a different search space under the same evolutionary search. Overall, we highlight that our search results on `einspace` are competitive, even with far weaker search space priors.

One dataset where our searches struggle is CIFARTile, a more complex version of the CIFAR10 dataset. While large improvements are made to the baseline networks, they still lag behind other NAS methods. This shows how the strong and restricted focus on ConvNets within the DARTS search space is highly successful for traditional computer vision style tasks that have been common in the literature.

## 5 Limitations

Our search space, designed for diversity, is extremely large and uniquely unbounded in terms of depth and width. This complexity makes formulating one-shot methods like ENAS [36] or DARTS [30] challenging. Instead, developing an algorithm to learn the probabilities of the PCFG may be more feasible. This approach, however, must address the grammar's context-free nature where sampling probabilities do not consider network depth, feature shape, or previous decisions, although this could be mitigated by using the parameters outlined in Section 3.6. Due to the relatively slow nature of our evolutionary search strategy (see Table 9 in Appendix C.5), we believe that finding more efficient search strategies for expressive spaces like ours is an important and exciting direction for future work.

Another issue is ambiguity arising from the possibility of multiple derivation trees for a single architecture, primarily due to the multiple ways of stacking sequential modules. Moreover, we have found that through sampling and mutation, some architectures' components reduce to the `identity` operation, from e.g. stacked `identity` and `permute` operations within sequential and routing modules. Finding ways to represent the equivalence classes of derivation trees can thus be powerful for reducing effective search space size.

Finally, we designed `einspace` to be diverse, but some key structures cannot be represented in its current form. There are no options for recurrent computation, as found in RNNs and the new wave of state-space models like Mamba [17]. We believe this can be integrated via the inclusion of a recurrent module that repeats the computation of the components within it—however we leave more detailed exploration of this direction to future work. We also chose to keep the options for activations and normalisation layers as small as possible since in practice the benefit from changing these is minor.

## 6 Conclusion

We have introduced `einspace`: an expressive NAS search space based on a parameterised PCFG. We show that our work enables the construction of a comprehensive and diverse range of existing state-of-the-art architectures and can further facilitate discovery of novel architectures directly from fundamental operations. With only simple search strategies, we report competitive resulting architectures across a diverse set of tasks, highlighting the potential value of defining highly expressive search spaces. We further demonstrate the utility of initialising search with existing architectures as priors. We believe that future work on developing intelligent search strategies within `einspace` can lead to exciting advancements in neural architectures.

## Acknowledgements and Disclosure of Funding

The authors are grateful to Joseph Mellor, Henry Gouk, and Thomas L. Lee for helpful suggestions. This work was supported by the Engineering and Physical Sciences Research Council [EP/X020703/1].

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

# A    Search Space Details

## A.1    Sampling

In this section, we explain the process of sampling an architecture from our parameterised PCFG through an example. We specifically focus on how the running example from the main paper, a simple convolutional block with a skip connection, could be generated.

We begin the process with the starting symbol S, which could produce several forms based on the production rules, including ( M M ), ( B M A ), or ( P M R ). Since our block includes a skip connection, the macro structure of our architecture is best represented by a branching module ( B M A ).

Within this module, we expand the string from left to right, thereby starting with ( B ). The specific branching operation that fits our goal is ( B → clone ) as we wish to later combine a transformed version of the tensor with itself. Since we have two branches, they are produced separately (see branching prior) and our module becomes ( B $M_1 M_2$ A ).

For the first branch, ( $M_1$ ), we need a set of components that constitute a convolution followed by batch normalisation and an activation. Since this involves several composed operations we first expand into a sequential module ( $M_1 \rightarrow M_3 M_4$ ). The first of these operations represents the convolution. Within einspace, a convolution, conv($x$), is decomposed into the three operations, col2im(linear(im2col($x$))). Thus, in our grammar we unfold it as a routing module, ( $M_3 \rightarrow P M_5 R$ ) which further produces ( P → im2col ), ( $M_5 \rightarrow$ C → linear ) and ( R → col2im ). The normalisation and activation are then generated under ( $M_4$ ), defined as another sequential module ( $M_4 \rightarrow M_6 M_7$ ) with ( $M_6 \rightarrow$ C → norm ) and ( $M_7 \rightarrow$ C → relu ). The second branch, $M_2$, acts as a skip connection and is thus derived as ( $M_2 \rightarrow$ C → identity ).

To finalise the architecture, the aggregation symbol ( A ) merges the tensors back into one unit. To complete the residual connection, we use the rule ( A → add ).

The full derivation tree in this example is shown in Figure 3 of the main paper. This general sampling process allows the creation of complex neural network architectures from a structured and interpretable set of rules.

## A.2    Fundamental Operations

Our grammar in the main paper is somewhat simplified. There are some fundamental operations that have hyperparameters that allow multiple versions to be chosen. They are detailed here.

**Branching functions**. For the production rule ( B → clone | group ), the full set of options is:

$$
\begin{aligned}
\text{B} \quad \rightarrow \quad & \texttt{clone(b=2)} \,|\, \texttt{clone(b=4)} \,|\, \texttt{clone(b=8)} \,| & (10) \\
& \texttt{group(dim=1,b=2)} | \texttt{group(dim=1,b=4)} | \texttt{group(dim=1,b=8)}| & (11) \\
& \texttt{group(dim=2,b=2)} | \texttt{group(dim=2,b=4)} | \texttt{group(dim=2,b=8)}| & (12) \\
& \texttt{group(dim=3,b=2)} | \texttt{group(dim=3,b=4)} | \texttt{group(dim=3,b=8)}, & (13)
\end{aligned}
$$

where b refers to the branching factor and dim is the dimension we group over.

**Aggregation functions**. Similarly, for the production rule ( A → matmul | add | concat ), the full set of options is:

$$
\begin{aligned}
\text{A} \quad \rightarrow \quad & \texttt{matmul(scaled=False)} \,|\, \texttt{matmul(scaled=True)} \,|\, \texttt{add} \,| & (14) \\
& \texttt{concat(dim=1,b=2)} | \texttt{concat(dim=1,b=4)} | \texttt{concat(dim=1,b=8)}| & (15) \\
& \texttt{concat(dim=2,b=2)} | \texttt{concat(dim=2,b=4)} | \texttt{concat(dim=2,b=8)}| & (16) \\
& \texttt{concat(dim=3,b=2)} | \texttt{concat(dim=3,b=4)} | \texttt{concat(dim=3,b=8)}, & (17)
\end{aligned}
$$

where scaled=True makes the matmul operation equivalent to the scaled dot product used in many transformer architectures, dim is the dimension we concatenate over and b is the branching factor.

**Routing functions**. The im2col and col2im functions are implemented to offer the standard functionality that enables convolutional operations, including variables that set the kernel sizes, stride, dilation and padding. Below are the full set of options for im2col. The col2im only takes the predicted output

shape as a parameter so we can include only a single version of this operation.

$$\texttt{im2col(k=1,s=1,p=0)}, \quad \texttt{im2col(k=1,s=2,p=0)}, \tag{18}$$

$$\texttt{im2col(k=3,s=1,p=1)}, \quad \texttt{im2col(k=3,s=2,p=1)}, \tag{19}$$

$$\texttt{im2col(k=4,s=4,p=0)}, \quad \texttt{im2col(k=8,s=8,p=0)}, \quad \texttt{im2col(k=16,s=16,p=0)}, \tag{20}$$

where k is the kernel size, s is the stride and p the padding.

For the permute operation, there are six versions of the order parameter $o$. There is one for the Col mode and five for the Im mode:

$$\texttt{permute(o=(2,1))}, \tag{21}$$

$$\texttt{permute(o=(1,3,2))}, \quad \texttt{permute(o=(2,1,3))}, \quad \texttt{permute(o=(2,3,1))}, \tag{22}$$

$$\texttt{permute(o=(3,1,2))}, \quad \texttt{permute(o=(3,2,1))}. \tag{23}$$

For completeness, the identity operation can also be included, making two versions in the Col mode (with identity=permute(o=(1,2))) and six in the Im mode (with identity=permute(o=(1,2,3))).

**Computation functions**. For linear layers, we vary the output dimension $d$ across powers of two:

$$\texttt{linear(d=16)}, \quad \texttt{linear(d=32)}, \quad \texttt{linear(d=64)}, \tag{24}$$

$$\texttt{linear(d=128)}, \quad \texttt{linear(d=256)}, \quad \texttt{linear(d=512)}, \tag{25}$$

$$\texttt{linear(d=1024)}, \quad \texttt{linear(d=2048)}. \tag{26}$$

The norm operation takes on the batch-norm functionality in the Im mode and layer-norm in Col mode. The softmax is just a softmax operation applied to the final dimension, the relu activation is implemented as the single option leaky-relu and pos-enc is a learnable positional encoding.

## A.3 Patch Embeddings and Convolutions

In this section we provide some more information about how the routing module can represent common components.

The routing module, ( M $\rightarrow$ P M R ), puts a prior on certain types of operations inside our architectures. A patch embedding, such as those found in many transformers, is achieved by the following derivation: ( M $\rightarrow$ im2col linear identity ), while a convolution can be obtained by ( M $\rightarrow$ im2col linear col2im ). In terms of the process required to sample such combinations, the first is easy since there are no dependencies between the operations. The second, however, is more complicated and requires some discussion.

Let $x$ be a tensor in $\mathbb{R}^{3\times32\times32}$ and let us consider a routing module containing the functions: im2col, linear, col2im. The functions will be applied in order to the input tensor, giving us

$$x' = \texttt{im2col}(x), \tag{27}$$

$$x'' = \texttt{linear}(x'), \tag{28}$$

$$y = \texttt{col2im}(x''). \tag{29}$$

The shapes of the intermediate and final tensors $\{x', x'', y\}$ depend on several function hyperparameters. These are listed below.

Table 2: Hyperparameters for the three functions involved in a convolution component.

| im2col | col2im | linear |
|---|---|---|
| $k_{in} = 7$ (kernel size) | $k_{out} = 7$ (kernel size) | $c_{out} = 64$ (output channels) |
| $s_{in} = 2$ (stride) | $s_{out} = 2$ (stride) | |
| $p_{in} = 3$ (padding) | $p_{out} = 0$ (padding) | |

The input tensor has height dimension $h_{in}$ and width $w_{in}$. The im2col operation will extract column vectors from this space a number of times depending on the kernel size $k_{in}$, stride $s_{in}$ and padding $p_{in}$

values in the table above. The number of column vectors equals

$$l = \left\lfloor \frac{h_{in} + 2 \times p_{in} - (k_{in} - 1) - 1}{s_{in}} + 1 \right\rfloor \times \left\lfloor \frac{w_{in} + 2 \times p_{in} - (k_{in} - 1) - 1}{s_{in}} + 1 \right\rfloor, \tag{30}$$

which in our case gives $l = 256$. The shapes of all intermediate tensors can therefore be written as in Table 3.

Table 3: Tensor shapes in the forward pass of our convolution component; $c_{in} = 3, h_{in} = 32, w_{in} = 32$ in this example.

| Tensor | Shape |
|--------|-------|
| $x$ | $[c_{in}, h_{in}, w_{in}]$ |
| $x'$ | $[l, c_{in} \times k_{in} \times k_{in}]$ |
| $x''$ | $[l, c_{out}]$ |
| $y$ | $[c_{out}, h_{out}, w_{out}]$ |

Therefore, to successfully apply the `col2im` function, the constraint $l = h_{out} \times w_{out}$ must be satisfied. From Equation 30 we can see that the output height and width can be predicted by the `im2col` parameters

$$h_{out} = \left\lfloor \frac{h_{in} + 2 \times p_{in} - (k_{in} - 1) - 1}{s_{in}} + 1 \right\rfloor, \tag{31}$$

$$w_{out} = \left\lfloor \frac{w_{in} + 2 \times p_{in} - (k_{in} - 1) - 1}{s_{in}} + 1 \right\rfloor. \tag{32}$$

So, in practice, the `im2col` operation fully defines the behaviour of the convolution—apart from the number of output channels defined by the linear layer—and the col2im only rearranges the tensor back into its correct 3D form. This is trivial in the case where $h_{in} = w_{in}$ since $h_{out} = w_{out} = \sqrt{l}$. However, if $h_{in} \neq w_{in}$, then the predicted output shapes must be remembered until the `col2im` operation is applied.

Thus, in our sampling and mutation algorithm, whenever an `im2col` operation is sampled, we must store the predicted output shape until a corresponding `col2im` is applied. Additionally, the dimensionality of $l$ must not change in the connecting branch as it would break the constraint $l = h_{out} \times w_{out}$.

### A.4 Branching Rate of the CFG

If a PCFG is consistent, the probabilities of all finite derivations sum to one, or equivalently, it has a zero probability of generating infinite strings or derivations [8]. For us, that means a sampled architecture can not be infinitely large, and that the sampling algorithm will halt with probability one. In order to check if a CFG is consistent, we can inspect the spectral radius $\rho$ of its non-terminal expectation matrix [8]. If $\rho < 1$, then the PCFG is consistent. This expectation matrix is indexed by the non-terminals in the grammar (both the columns and the rows), and at each cell it provides the expected number of instances the column non-terminal being generated from the row non-terminal by summing the probabilities of the row non-terminal multiplied by the count of the column non-terminal in each rule.

We can also solve a (simple, in our case) set of linear equations in order to compute the expected length of an architecture string, $\ell$, as a function of the rule probabilities. More specifically, denote by $\mathbb{E}[A]$ the expected length of string generated by a nonterminal in the grammar $A$. Then $\ell = \mathbb{E}[\text{S}]$, where:

$$\mathbb{E}[\text{S}] = \sum_{\text{S} \to \alpha} p(\text{S} \to \alpha \mid \text{S}) \sum_i \mathbb{E}[\alpha_i] \tag{33}$$

$$\mathbb{E}[\text{M}] = \sum_{\text{M} \to \alpha} p(\text{M} \to \alpha \mid \text{M}) \sum_i \mathbb{E}[\alpha_i] \tag{34}$$

$$\mathbb{E}[\text{B}] = 1 \tag{35}$$

$$\mathbb{E}[\text{A}] = 1 \tag{36}$$

$$\mathbb{E}[\text{P}] = 1 \tag{37}$$

$$\mathbb{E}[\text{R}] = 1 \tag{38}$$

$$\mathbb{E}[\text{C}] = 1 \tag{39}$$

In the above, $\alpha$ is the right hand side of a rule and $\alpha_i$ varies over the nonterminals in that right hand side.

## A.5    Adjustments for One-Dimensional Tasks

We have presented our new search space primarily with the application to two-dimensional data in mind—where, confusingly, the input tensor is of the shape $(\texttt{C, H, W})$, i.e. two spatial dimensions and one channel dimension. In order to search for architectures on 1D datasets—with input tensors of shape $(\texttt{S, D})$, i.e. one sequence dimension and one token dimension—we need to adjust the `einspace` CFG to make it compatible.

We replace the routing functions `im2col` and `col2im` with operations that perform the full 1D convolution directly—without decomposing the operation—and place them into the computation function group. Below are the new `conv1d` functions in the 1D variant of `einspace`

$$\texttt{conv1d(k=1,s=1,p=0,d=32)}, \quad \texttt{conv1d(k=1,s=1,p=0,d=64)} \tag{40}$$
$$\texttt{conv1d(k=1,s=1,p=0,d=128)}, \quad \texttt{conv1d(k=1,s=1,p=0,d=256)} \tag{41}$$
$$\texttt{conv1d(k=3,s=1,p=1,d=32)}, \quad \texttt{conv1d(k=3,s=1,p=1,d=64)} \tag{42}$$
$$\texttt{conv1d(k=3,s=1,p=1,d=128)}, \quad \texttt{conv1d(k=3,s=1,p=1,d=256)} \tag{43}$$
$$\texttt{conv1d(k=5,s=1,p=2,d=32)}, \quad \texttt{conv1d(k=5,s=1,p=2,d=64)} \tag{44}$$
$$\texttt{conv1d(k=5,s=1,p=2,d=128)}, \quad \texttt{conv1d(k=5,s=1,p=2,d=256)} \tag{45}$$
$$\texttt{conv1d(k=8,s=1,p=3,d=32)}, \quad \texttt{conv1d(k=8,s=1,p=3,d=64)} \tag{46}$$
$$\texttt{conv1d(k=8,s=1,p=3,d=128)}, \quad \texttt{conv1d(k=8,s=1,p=3,d=256)}. \tag{47}$$

Some versions of the `group`, `concat` and `permute` operations are also removed as they operate on a dimension that now doesn't exist. Below is the adjusted grammar for this variant.

```
S  →  M  M  |  B  M  A  |  R  M  R,
M  →  M  M  |  B  M  A  |  R  M  R  |  C,
B  →  clone  |  group,
A  →  matmul  |  add  |  concat,
R  →  identity  |  permute,
C  →  identity  |  conv1d  |  linear  |  norm  |  relu  |  softmax  |  pos-enc.
```

In the NASBench360 results presented in Table C.1, this 1D variant of `einspace` is used for the datasets ECG, Satellite and Deepsea. The baseline WideResNet-16-4 architecture is also adjusted to handle the 1D data, as described in [55].

## A.6    Size of the Search Space

In this section we will discuss the size of the introduced search space.

Our `einspace` grammar contains recursive rules, e.g. ($\texttt{M} \rightarrow \texttt{M M}$). Due to this recursion the grammar generates a language with an infinite number of strings. We know from Section A.4 that since our grammar is consistent, the architecture sampling process always terminates. This means every architecture derivation tree is finite in size. So, while `einspace` covers an infinite number of architectures, each such architecture is finite. Let the *architecture string* denote the left-to-right sequence of leaves of a derivation tree. We define $S_n$ to be the set of all architecture strings of length $n$. As we will show next, the size of $S_n$ grows exponentially with $n$. To do this, we first introduce a few new concepts.

A *Dyck language* [25] is a formal language consisting of strings that represent balanced and properly nested sequences of pairs of symbols, typically parentheses, where each opening symbol must have a corresponding closing symbol, and at no point in the string can the number of closing symbols exceed the number of opening symbols. A Dyck language can be generated by a context-free grammar defined over a set of terminal symbols $\Sigma = \{(,)\}$. Generalising this, Dyck-$k$ denotes a Dyck language with $k$ distinct pairs of matching symbols, e.g. Dyck-2 has $\Sigma = \{(,),[,]\}$. The growth of a Dyck-1 language is described by the Catalan numbers, reflecting the exponential increase in valid strings as the length of the input increases, i.e. the number of strings with $m$ matching pairs of symbols is the $m$th Catalan number

$C_m = \frac{1}{m+1}\binom{2m}{m}$. Asymptotically, the number of such strings grows as $O(4^m)$. Dyck-2 languages do not follow the Catalan numbers but still grow as $O(4^m)$.

We can rewrite (and somewhat simplify) our `einspace` CFG from Section 3.3 in order to generate a Dyck language

$$
\begin{aligned}
\texttt{S} &\rightarrow \texttt{M M | ( M ) | [ M ]}, \\
\texttt{M} &\rightarrow \texttt{M M | ( M ) | [ M ] | C}, \\
\texttt{C} &\rightarrow \texttt{1 | 2 | 3 | ...}
\end{aligned}
$$

The set of non-terminal symbols {B, A, P, R} has been replaced with the terminal set of parentheses and square brackets {(, ), [, ]} and, for simplicity, the terminal symbols of C have been replaced by integers. This makes it a Dyck-2 language with the addition of the C symbol that can be expanded into several terminal options.

While a normal Dyck-2 language grows as $O(4^m)$, ours is more complex. Firstly, each non-terminal symbol M will eventually produce a C which leads to a terminal. Let the number of terminals derivable from C be $\chi$, and the number of occurrences of the symbol C in the derivation of an architecture be $c$. The number of possible strings we can obtain from just the C symbols thus grows as $O(\chi^c)$. Secondly, the brackets we introduced can, in our original CFG, themselves be derived into terminal symbols for branching, aggregation and routing functions via {B, A, P, R}. Let the maximum number of terminals derivable from any of these be $\beta$. We already know that the number of balanced brackets is $m$, meaning the total number of terminals coming from these is $2m$. Therefore, the number of architecture strings in $S_n$ grows as

$$O(4^m \cdot \beta^{2m} \cdot \chi^c), \tag{48}$$

where $m$ is the combined number of branching and routing modules, $c$ is the number of computation functions, and $n = 2m + c$. Some architectures will contain many nested branching or routing functions and be dominated by the first two factors, and some will contain many sequential modules and computation functions and be dominated by the third.

### A.7 Comparison to Other Search Spaces

In Table 4 we compare `einspace` to other popular search spaces in the literature along several axes. We highlight some of the most important differences here:

- `einspace` uniquely unifies multiple architectural families (ConvNets, Transformers, MLP-only) into one single expressive space while the CFG-based framework of Schrodi et al. [46] has variations of spaces centred around ConvNets only, and a separate instantiation focusing only on Transformers.

- `einspace` extends to probabilistic CFGs. This enables a set of benefits that include (i) allowing experts to define priors on the search space via probabilities, and (ii) enabling a broader range of search algorithms that incorporate uncertainty estimates inside the search itself.

- `einspace` contains recursive production rules (e.g. $M \rightarrow MM$), meaning the same rule can be expanded over and over again, providing an infinite space and a very high flexibility in macro structures. In contrast, [46] instead focuses on fixed hierarchical levels that limits the macro structure to a predefined—though very large—set of choices.

- `einspace` encodes architectures in the form of derivation trees. These allow for mutations that can effectively alter both the macro structure and the individual components of an architecture. Modifications of this class are more difficult if using e.g. the more rigid graph encodings.

Overall, we highlight that our experimental search results on `einspace` are competitive with previous work, even with far weaker priors on the search space.

Table 4: Comparison of `einspace` with existing search spaces. RS: random search, RE: regularised evolution, RL: reinforcement learning, BO: Bayesian optimisation and GB: gradient-based. ‡ The size of `einspace` is infinite, but we can bound the number of possible architectures strings of a certain length, as discussed in Section A.6. † Gradient-based search is difficult in these spaces due to their size, but other weight-sharing methods may be available. * The paper introducing hNASBench201 [46] also considers versions of the search space for Transformer language models.

| | | Search Space Properties | | Available Search Strategies | | | | |
| | Type | Size | Focus | RS | RE | RL | BO | GB |
|---|---|---|---|---|---|---|---|---|
| `einspace` | PCFG | Infinite$^{\ddagger}$ | ConvNets, Transformers, MLP-only | ✓ | ✓ | ✓ | ✓ | † |
| hNASBench201 | CFG | $10^{446}$ | ConvNets* | ✓ | ✓ | ✓ | ✓ | † |
| NASBench201 | Cell | $10^4$ | ConvNets | ✓ | ✓ | ✓ | ✓ | ✓ |
| NASBench101 | Cell | $10^5$ | ConvNets | ✓ | ✓ | ✓ | ✓ | ✓ |
| DARTS | Cell | $10^{18}$ | ConvNets | ✓ | ✓ | ✓ | ✓ | ✓ |

# B   Implementation and Experimental Details

## B.1   Networks

**Baseline networks**
We use convolutional baselines of ResNet18 [19] and WideResNet-16-4 [64]. Both stems use a $3 \times 3$ convolution instead of the standard $7 \times 7$ as most input shapes in the datasets we use are small. The former contains a max-pooling layer in the stem, which for simplicity we decide to not represent in our search space. Instead we modify the pooling operation and replace it with a $3 \times 3$ convolution with stride 2. This has been shown to be equally powerful [49] and in our experiments we find that it performs similarly. Our ViT model is a small 4-layer network with a patch size of 4, model width of 512, and 4 heads. The MLP-Mixer shares the same patch embedding stem with a patch size of 4. It has 8 layers and, similarly, a model width of 512. The channel mixer expands the dimension by 4 and the token mixer cuts it in half. The models have the following number of parameters (given an input image of shape [3, 64, 64]): Resnet18: 11.2M, WRN16-4: 2.8M, ViT: 4.4M and MLP-Mixer: 6.5M.

**Network head**
Every network that is instantiated contains a few common operations. For classification tasks, the network head takes the following form: the output features of the sampled backbone are reduced to their channel dimension via `reduce(x, 'B C H W -> B C', 'mean')`[4] or `reduce(x, 'B C H -> B C', 'mean')`, depending on if the input features `x` are in `Im` or `Col` mode. Second, a final linear layer that maps the channel dimension to the target dimension (i.e., the number of classes) is appended. For dense prediction tasks: the head contains an adaptive average pooling layer that upsamples the two final dimensions of the backbone features to the target image size. If the features are in the `Col` mode, we insert a new dimension after the batch size. Then regardless of mode, a linear layer adjusts the number of channels to the target channel number.

**Network training**
Each network is trained and evaluated separately with no weight sharing. During the search phase we minimise the loss on a train split and compute the validation metric on a validation set. To evaluate the final chosen module, we retrain on train+val splits and evaluate on test. To speed up our experiments, the inner loop optimisation of architectures uses fewer epochs compared to the evaluation phase. All networks are trained using the SGD optimizer with momentum of 0.9. The values for learning rate, weight decay, batch size and more can be found in Table 5.

## B.2   Datasets

Our experimental evaluation covers 19 different datasets, with sizes ranging from thousands of data points, to a million (Satellite from NASBench360), and spatial resolutions of up to $256 \times 256$ (Cosmic from NASBench360). In this section we briefly describe these parts in detail.

We followed the official instructions of Unseen NAS [16] to setup the datasets. Descriptions of each dataset follow:

---

[4]The notation used here comes from the Python package `einops` [44], which implements a `reduce` function.

Table 5: Hyperparameters for each dataset, taken from Geada et al. [16] for Unseen NAS and Tu et al. [55] for NASBench360. We set the number of search epochs to be one eighth of the evaluation epochs to speed up the search stage without significantly compromising on signal quality. For the Unseen NAS datasets (top set) we report the accuracy across all datasets. For NASBench360 (second set) and CIFAR10 (bottom), the metrics differ and we report the 0-1 error instead of accuracy to align with the other metrics focused on error minimisation.

| Dataset name | Metric type | Baseline model | Epochs (search) | Epochs (eval) | Batch size | Learning rate | Weight decay | Mom-entum |
|---|---|---|---|---|---|---|---|---|
| AddNIST | Accuracy | ResNet18 | 8 | 64 | 256 | 0.04 | $3\times10^{-4}$ | 0.9 |
| Language | Accuracy | ResNet18 | 8 | 64 | 256 | 0.04 | $3\times10^{-4}$ | 0.9 |
| MultNIST | Accuracy | ResNet18 | 8 | 64 | 256 | 0.04 | $3\times10^{-4}$ | 0.9 |
| CIFARTile | Accuracy | ResNet18 | 8 | 64 | 256 | 0.04 | $3\times10^{-4}$ | 0.9 |
| Gutenberg | Accuracy | ResNet18 | 8 | 64 | 256 | 0.04 | $3\times10^{-4}$ | 0.9 |
| Isabella | Accuracy | ResNet18 | 8 | 64 | 256 | 0.04 | $3\times10^{-4}$ | 0.9 |
| GeoClassing | Accuracy | ResNet18 | 8 | 64 | 256 | 0.04 | $3\times10^{-4}$ | 0.9 |
| Chesseract | Accuracy | ResNet18 | 8 | 64 | 256 | 0.04 | $3\times10^{-4}$ | 0.9 |
| CIFAR100 | 0-1 error | WRN16-4 | 25 | 200 | 128 | 0.1 | $5\times10^{-4}$ | 0.9 |
| Spherical | 0-1 error | WRN16-4 | 25 | 200 | 128 | 0.1 | $5\times10^{-4}$ | 0.9 |
| NinaPro | 0-1 error | WRN16-4 | 25 | 200 | 128 | 0.1 | $5\times10^{-4}$ | 0.9 |
| FSD50K | 1 - mAP | WRN16-4 | 25 | 200 | 256 | 0.1 | $5\times10^{-4}$ | 0.9 |
| Darcy Flow | relative $\ell_2$ | WRN16-4 | 25 | 200 | 4 | 0.001 | $5\times10^{-4}$ | 0.9 |
| PSICOV | $MAE_8$ | WRN16-4 | 25 | 200 | 8 | 0.001 | $5\times10^{-4}$ | 0.9 |
| Cosmic | 1 - AUROC | WRN16-4 | 25 | 200 | 8 | 0.001 | $5\times10^{-4}$ | 0.9 |
| ECG | 1 - F1 | WRN16-4 | 25 | 200 | 256 | 0.1 | $5\times10^{-4}$ | 0.9 |
| Satellite | 0-1 error | WRN16-4 | 25 | 200 | 4096 | 0.1 | $5\times10^{-4}$ | 0.9 |
| Deepsea | 1 - AUROC | WRN16-4 | 25 | 200 | 256 | 0.1 | $5\times10^{-4}$ | 0.9 |
| CIFAR10 | 0-1 error | WRN16-4 | 25 | 200 | 128 | 0.1 | $5\times10^{-4}$ | 0.9 |

**AddNIST**

This dataset is derived from the MNIST dataset. Specifically, each RGB image is computed by stacking three MNIST images in the channel dimension. Each image has the shape $3\times28\times28$. It has a total of 20 categories; the class label is computed by summing the MNIST labels in all three channels. Among the 70,000 images, 45,000 are used for training, 15,000 are used for validation, and 10,000 images are used for testing.

**Language**

Language consists of six-letter words extracted from dictionaries of ten Latin alphabet languages: English, Dutch, German, Spanish, French, Portuguese, Swedish, Zulu, Swahili, and Finnish. Words containing diacritics or the letters 'y' and 'z' are excluded, making an alphabet of 24 letters. Each sample consists of four words encoded into a binary image of shape $1\times24\times24$. The task is to predict the language of the sample. Along the y-axis are the letter positions in the concatenated 24-letter string, and along the x-axis are the letters in the alphabet. As an example, a one in the position (0, 0) indicates that the first letter in the string is an 'a'. The dataset is split into 50,000 training samples, 10,000 validation samples, and 10,000 test samples.

**MultNIST**

MultNIST is a dataset designed similarly to AddNIST, originating from the same research objective. Each channel of the 3 channel images contains an image from the MNIST dataset. Each image has the shape $3\times28\times28$. The dataset is divided into 50,000 training images, 10,000 validation images, and 10,000 test images. Unlike AddNIST, MultNIST comprises ten classes (0-9), the label for each MultNIST image is computed using the formula $l=(r\times g\times b)\bmod 10$, where $r$, $g$ and $b$ are the MNIST labels of the red, green, and blue channels, respectively.

**CIFARTile**

CIFARTile is a dataset where each image is a composite of four CIFAR-10 images arranged in a $2\times2$ grid, making each sample an image of shape $3\times64\times64$. The dataset is divided into 45,000

training images, 15,000 validation images, and 10,000 test images. CIFARTile has four classes (0-3), determined by the number of distinct CIFAR-10 classes in each grid, minus one.

### Gutenberg

Gutenberg is a dataset sourced from Project Gutenberg. It includes texts from six authors, with basic text preprocessing applied: punctuation removal, diacritic conversion, and elimination of structural words. The dataset contains consecutive sequences of three words (3-6 letters each), padded to 6 characters and concatenated into 18-character strings. These strings are converted into images with size $1 \times 27 \times 18$, with the x-axis representing character positions and the y-axis representing alphabetical letters or spaces. The task is to predict the author of each sequence. The dataset is split into 45,000 training, 15,000 validation, and 6,000 test images.

### Isabella

Isabella is a dataset derived from musical recordings of the Isabella Stewart Gardner Museum, Boston. It includes four classes based on the era of composition: Baroque, Classical, Romantic, and 20th Century. The recordings are split into five-second snippets and converted into 64-band spectrograms, resulting in images with dimensions $1 \times 64 \times 128$. The dataset is divided into 50,000 training images, 10,000 validation images, and 10,000 test images, ensuring no overlap of recordings across splits. The task is to predict the era of composition from the spectrogram.

### GeoClassing

GeoClassing is based on the BigEarthNet dataset. It uses satellite images initially labeled for ground-cover classification but reclassified by the European country they depict. The dataset includes ten classes: Austria, Belgium, Finland, Ireland, Kosovo, Lithuania, Luxembourg, Portugal, Serbia, and Switzerland. Each image is of size $3 \times 64 \times 64$. The dataset is split into 43,821 training images, 8,758 validation images, and 8,751 test images. The task is to predict the country depicted in each image based on topology and ground coverage.

### Chesseract

Chesseract is a dataset derived from public chess games of eight grandmasters. The dataset consists of the final 15% of board states from these games. Each position is one-hot encoded by piece type and color, resulting in $1 \times 8 \times 8$ images. The dataset is divided into 49,998 training images, 9,999 validation images, and 9,999 test images, ensuring no positions from the same game appear across splits. Each image is classified into one of three classes: White wins, Draw, or Black wins. The task is to predict the game's result based on the given board position. We pad the input with 5 zero-valued pixels to make a $12 \times 18 \times 18$ tensor.

We follow the official instructions of NASBench360 [55] to setup the datasets. Dataset descriptions follow:

### CIFAR100

CIFAR100 is a widely known image classification dataset with 100 fine-grained classes. Each image is of size $3 \times 32 \times 32$. The dataset is split into 40,000 training, 10,000 validation and 10,000 testing images. We preprocess the images by applying random crops, horizontal flips, and normalisation.

### Spherical

Spherical dataset consists on classifying spherically projected CIFAR100 images. Specifically, CIFAR images are projected to the northern hemisphere with a random rotation. Each image is of size $3 \times 60 \times 60$. Spherical dataset uses the same split ratios as CIFAR100. In this case, there is no data augmentation nor pre-processing steps.

### NinaPro

NinaPro is a dataset for classifying hand gestures given their electromyography signals. EMG data signals are collected with two Myo armbands as wave signals. Wave signals, along with their wavelength and frequency, are processed 2D signals of shape $1 \times 16 \times 52$. There are 18 classes, which are heavily imbalanced, with the neutral position amounting for 65% of all gestures. The dataset is split into 2,638 training samples, 659 validation samples, and 659 testing samples. No further data augmentation is applied.

### FSD50K

Freesound Dataset 50k (FSD50K) is a collection of 51,197 sound clips, categorised into 200

categories, where each clip can receive multiple labels. The task is to classify the sound event from its log-mel spectrogram, and the performance is computed via the mean average precision (mAP). We resample the raw audio files at a frequency of 22,050 Hz and convert them into 96-band log-mel spectrograms. From these longer audio files, we extract shorter, overlapping 1-second segments, resulting in an input size of $1 \times 101 \times 96$. During training, one randomly-selected segment per clip is used, rather than all segments. Background noise is added to 75% of the training data as part of data augmentation. The validation set consists of $4,170$ clips and the test set of 10,231 clips.

### Darcy Flow
Darcy Flow is a regression task for predicting the solution of a 2D PDE at a predefined later stage given some 2D initial conditions. The input is a $1 \times 85 \times 85$ image describing the initial state of the fluid. The output should match the same dimensions. The dataset is split into 900 images for training, 100 for validation, and 100 for test. Input data is normalised.

### PSICOV
This dataset concerns the use of neural networks in the protein folding prediction pipeline. It uses proteins from one source, DeepCov, for training and validation. DeepCov contains 3,456 proteins each with shape $57 \times 128 \times 128$. The validation set consists of 100 proteins, with the rest used for training. The test set comes from another source, PSICOV, with 150 proteins. These proteins come in features of a different shape, $1 \times 512 \times 512$. Due to this, the evaluated network takes each non-overlapping $1 \times 128 \times 128$ patch as input. The labels represent pairwise distances between residues. The evaluation metric is mean absolute error (MAE) computed on distances below 8Å, denoted as $MAE_8$.

### Cosmic
Cosmic contains images from the F435W filter collected from the Hubble Space Telescope. It aims to identify cosmic rays (corrupted pixels) in the images. Inputs are images of $1 \times 256 \times 256$, and outputs are pixel binary predictions (artifact vs. no-artifact). The dataset is split into 4,347 images for training, 483 for validation, and 420 for test.

### ECG
The ECG task concerns predicting irregularities in electrocardiograms. ECG recordings of 9-60 seconds are sampled at 300 Hz and labeled using four classes: normal, disease, other, or noisy rhythms. We process each recording with a fixed sliding window of 1,000 ms and stride of 500 ms. This transforms 2,186 single lead recordings into more than 330,000 segments. We use 261,740 of these for training, 33,281 for validation, and 33,494 for test. Each segment has the shape $1 \times 1,000$. The evaluation metric is the F1-score.

### Satellite
The goal of the Satellite task is to classify land cover maps for geo-surveying, for one million data points across 24 categories. Each data point is a single-channel satellite time-series of shape $1 \times 46$, with standard normalisation augmentation applied. The data is split with 800,000 samples for the training set, 100,000 for validation, and 100,000 for test.

### Deepsea
This dataset focuses on predicting the behaviour of chromatin proteins to aid in understanding genetic diseases. Each data point is a genome sequence of 1,000-base pairs (A, C, T or G), represented as a binary matrix of shape $1000 \times 4$, categorised across 36 classes of chromatin features. The training set contains 71,753 data points, with 2,490 for validation and 149,400 for testing. The evaluation metric is the area under the receiver operating characteristic (AUROC).

Finally, we also report results on the classic CIFAR10 dataset:

### CIFAR10
CIFAR10 is a widely known image classification dataset with 10 classes. Each image is of size $3 \times 32 \times 32$. The dataset is split into 40,000 training, 10,000 validation and 10,000 testing images. We preprocess the images by applying random crops, horizontal flips, and normalisation.

## B.3   Compute Resources

All our experiments ran on our two internal clusters with the following infrastructure:

Table 6: Lower is better. Our results on NASBench360 [55], where we report errors across all datasets as described in Table 5. We compare to the results from [55], where the GAEA algorithm on the DARTS search space is used, along with a human-designed expert architecture per dataset (Expert). RE(WRN) refers to initialising the regularised evolution search algorithm with the WideResNet16-4 (WRN) architecture. Note that we could not reproduce some baseline WRN performances in our code. We have therefore reported both the WRN from NB360 and our own WRN results. We use our own results for computing average ranks. **Best** and second best performance per dataset (excluding WRN from NB360).

| | WRN NB360 | WRN ours | GAEA DARTS | Expert | RE(WRN) einspace |
|---|---|---|---|---|---|
| CIFAR100 | 23.35 | 25.61 | 24.02 | **19.39** | 21.47 |
| Spherical | 85.77 | 76.32 | **48.23** | 67.41 | 66.37 |
| NinaPro | 6.78 | 10.32 | 17.67 | 8.73 | **6.37** |
| FSD50K | 0.92 | 0.92 | 0.94 | **0.62** | 0.65 |
| Darcy Flow | 0.073 | 0.032 | 0.026 | **0.008** | 0.014 |
| PSICOV | 3.84 | 5.71 | **2.94** | 3.35 | 4.38 |
| Cosmic | 0.245 | 0.245 | 0.229 | **0.127** | 0.730 |
| ECG | 0.43 | 0.59 | 0.34 | **0.28** | 0.46 |
| Satellite | 15.49 | 15.29 | **12.51** | 19.80 | 12.55 |
| DeepSea | 0.40 | 0.45 | 0.36 | **0.30** | 0.36 |
| Average rank ↓ | - | 3.60 | 2.35 | **1.70** | 2.35 |

Table 7: Higher is better. Accuracies for searches with DenseNet121 as well as finetuning pretrained networks on the Unseen NAS benchmark. First two column are the baseline performances of training ResNet18 and DenseNet121 from scratch. Next two columns are results when initialising our regularised evolution with ResNet18 and DenseNet121. Final two columns are results when finetuning ResNet18 and EfficientNet-B0 from their pretrained ImageNet weights.

| | | | RE | | Finetuning | |
|---|---|---|---|---|---|---|
| | RN18 | DN121 | RN18 | DN18 | RN18 | ENB0 |
| AddNIST | 93.36 | 94.72 | **97.54** | 94.84 | 94.69 | 94.77 |
| Language | 92.16 | 91.27 | 96.84 | **98.39** | 90.31 | 90.62 |
| MultNIST | 91.36 | 92.58 | **96.37** | 94.86 | 91.12 | 91.90 |
| CIFARTile | 47.13 | 56.37 | 60.65 | 66.06 | 52.26 | **79.32** |
| Gutenberg | 43.32 | 42.97 | **54.02** | 47.23 | 42.52 | 42.08 |
| Isabella | 63.65 | 43.65 | 64.30 | 42.89 | 62.35 | **67.46** |
| GeoClassing | 90.08 | 92.20 | 95.31 | 94.33 | 90.70 | **95.81** |
| Chesseract | 59.35 | 61.72 | 60.31 | 61.64 | 61.29 | 62.24 |
| Average acc. ↑ | 72.55 | 71.94 | **78.17** | 75.03 | 73.16 | 78.02 |

- AMD EPYC 7552 48-Core Processor with 1000GB RAM and 8× NVIDIA RTX A5500 with 24GB of memory
- AMD EPYC 7452 32-Core Processor with 400GB RAM and 7× NVIDIA A100 with 40GB of memory

Each experiment used a single GPU to train each architecture. Running 1000 iterations of RE(Scratch) on the quickest datasets (Language and Chesseract) took around 2 days, while the slowest (GeoClassing) took 4 days. We had very similar training times for RE(RN18) and RE(Mix) which ran for 500 iterations.

## C   Additional Results

### C.1   NASBench360

We test einspace on the diverse NASBench360 [55] to further showcase its potential. These results can be found in Table 6. In this setting the baseline network is a WideResNet16-4 (WRN)—which is

Table 8: Lower is better. Performance on CIFAR10 as measured by the 0-1 error.

|  | RN18 | RE(RN18) | RE(Mix) |
|---|---|---|---|
| CIFAR10 | 5.09 | **4.69** | 5.27 |

Table 9: Runtime of search algorithms, as well as the number of model parameters for the found architectures. Numbers for PC-DARTS on two datasets are missing due to missing logs from the authors of [16]. RE on `einspace` is the RE(Mix) variant, while RE on hNASBench201 searches from scratch.

| | | | AddNIST | Language | MultNIST | CIFARTile | Gutenberg | Isabella | GeoClassing | Chesseract |
|---|---|---|---|---|---|---|---|---|---|---|
| runtime | (hours) | DrNAS | 10 | 9 | 11 | 25 | 13 | 59 | 23 | 10 |
| | | PC-DARTS | 4 | - | 5 | 12 | 9 | 30 | - | 2 |
| | | RE(hNB201) | 15 | 72 | 21 | 59 | 9 | 59 | 37 | 9 |
| | | RE(einspace) | 55 | 71 | 32 | 62 | 42 | 80 | 65 | 42 |
| #params | ($\times 10^6$) | DrNAS | 4 | 4 | 5 | 3 | 3 | 4 | 4 | 4 |
| | | PC-DARTS | 3 | - | 3 | 3 | 2 | 2 | - | 2 |
| | | RE(hNB201) | 1 | 1 | 1 | 3 | 1 | 1 | 1 | 1 |
| | | RE(einspace) | 20 | 1 | 25 | 5 | 1 | 5 | 4 | 11 |

adapted for the 1D tasks ECG, Satellite and Deepsea as in [55]. We see that our regularised evolution with the baseline as the initial seed, RE(WRN), again consistently finds architectural improvements. It matches the performance of the GAEA search strategy on the DARTS search space, achieving the same average rank. On NinaPro, it even beats the Expert architecture, specifically designed for this task and to the best of our knowledge sets a new state-of-the-art. Note that we fail to exactly reproduce the WRN baseline network performance on several tasks, so we report both our own WRN values along with those from [55].

## C.2 More Baseline Architectures

We explore another baseline architecture, DenseNet121, and use it as the initial seed to our evolutionary search in `einspace`. In Table 7, we can see that the DenseNet121 on its own performs comparably to the ResNet18 model. When seeding search from the model with RE(DN121), we observe performance boosts similar to the gaps found between RN18 and RE(RN18), highlighting the general applicability of initialising search from different architectures within `einspace`.

## C.3 Finetuning

We present further results comparing our method against finetuning a model from pre-trained weights. In the experiments presented in Table 7, a finetuned EfficientNet-B0 matches or beats the ResNet18 baseline on the Unseen NAS datasets. However, RE(Mix) on `einspace` still often outperforms this method, e.g. on Gutenberg by 12%, although finetuning dominates on the CIFARTile vision task.

## C.4 CIFAR10

To complete our evaluation on common NAS benchmarks, we also report results on the CIFAR10 dataset in Table 8.

## C.5 Runtime and Parameter Count

Table 9 summarises the runtimes and parameter counts of architectures found using DrNAS, PC-DARTS, and RE on hNASBench201 and `einspace`. Notably, the gradient-based DrNAS and PC-DARTS demonstrate significantly shorter runtimes, with DrNAS achieving search times between 10 to 25 hours and PC-DARTS between 4 to 12 hours. In contrast, the black-box evolutionary methods show substantially longer runtimes, ranging from 9 to 80 hours, as they independently train a large number of networks. The search times for `einspace` are comparable to those for hNASBench201, though due to the potential increased complexity in candidate architectures, they can sometimes be significantly longer. In terms of model parameters, DrNAS and PC-DARTS consistently produce architectures

Table 10: The distribution of terminal and nonterminal symbols as well as average branching factors in 2000 sampled architectures with varying values for the computation module probability $p(\texttt{M}\rightarrow\texttt{C}|\texttt{M})$. For probability values 0.2 and 0.1 there is no data, as the sampling process is too time-consuming.

| Type | $p(\texttt{M}\rightarrow\texttt{C}|\texttt{M})$ | Min | Mean | Median | Std | Max |
|------|------|-----|------|--------|-----|-----|
| terminals | 0.9 | 2 | 3.87 | 3.00 | 2.84 | 28 |
| | 0.8 | 2 | 5.34 | 4.00 | 6.32 | 54 |
| | 0.7 | 2 | 6.72 | 4.00 | 11.22 | 118 |
| | 0.6 | 2 | 8.65 | 4.00 | 16.73 | 202 |
| | 0.5 | 2 | 40.22 | 5.00 | 303.65 | 4779 |
| | 0.4 | 2 | 100.15 | 8.00 | 749.61 | 12026 |
| | 0.3 | 2 | 6070.77 | 9.00 | 64863.45 | 878122 |
| non-terminals | 0.9 | 2 | 3.63 | 3.00 | 3.45 | 42 |
| | 0.8 | 2 | 5.04 | 3.00 | 6.91 | 64 |
| | 0.7 | 2 | 6.00 | 3.00 | 9.21 | 81 |
| | 0.6 | 2 | 7.74 | 4.00 | 13.61 | 165 |
| | 0.5 | 2 | 39.03 | 5.00 | 325.18 | 5219 |
| | 0.4 | 2 | 88.58 | 8.00 | 649.44 | 10417 |
| | 0.3 | 2 | 4588.13 | 8.00 | 50203.87 | 734497 |
| branching factor | 0.9 | 1 | 1.75 | 1.00 | 1.63 | 7.61 |
| | 0.8 | 1 | 2.05 | 1.00 | 1.95 | 7.73 |
| | 0.7 | 1 | 2.05 | 1.00 | 1.91 | 7.83 |
| | 0.6 | 1 | 2.17 | 1.00 | 1.96 | 7.73 |
| | 0.5 | 1 | 2.34 | 1.00 | 2.00 | 7.88 |
| | 0.4 | 1 | 2.81 | 1.75 | 2.14 | 7.87 |
| | 0.3 | 1 | 2.80 | 1.75 | 2.13 | 7.96 |

with parameter counts around 4 million, hNASBench201 is consistently more parameter efficient at around 1M, while `einspace` finds architectures that vary significantly in size from 1M–25M, showing flexibility in adaptation to task difficulty.

## C.6 Empirical Architecture Complexity

For our experiments we set the computation module probability to $p(\texttt{M} \rightarrow \texttt{C} \mid \texttt{M}) = 0.32$ using the branching rate method described in A.4. We next report empirical results for the architecture complexities as we vary this value. In Table 10 we can see that the complexity, as measured by the count of terminals and non-terminals in the derivation trees, grows as the probability decreases.

When searching for an architecture on a new unknown task, the flexibility of the search space is key. During our random searching on `einspace` we found that we sampled networks with parameter counts ranging from zero up to 1 billion, and from as few as two operations up to as many as 3,000. The frequency of all functions in `einspace` that appear in these networks can be found in Figure 7.

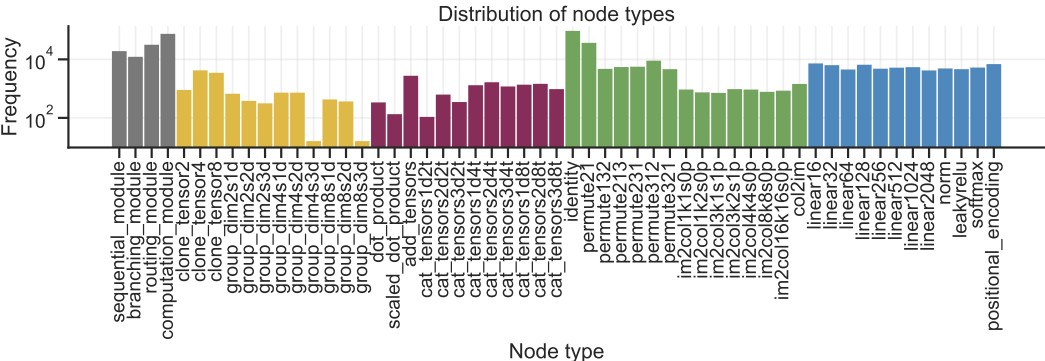

Figure 7: Frequency of each module/function in 8,000 sampled architectures with $p(\texttt{M} \rightarrow \texttt{C} \,|\, \texttt{M}) = 0.32$.

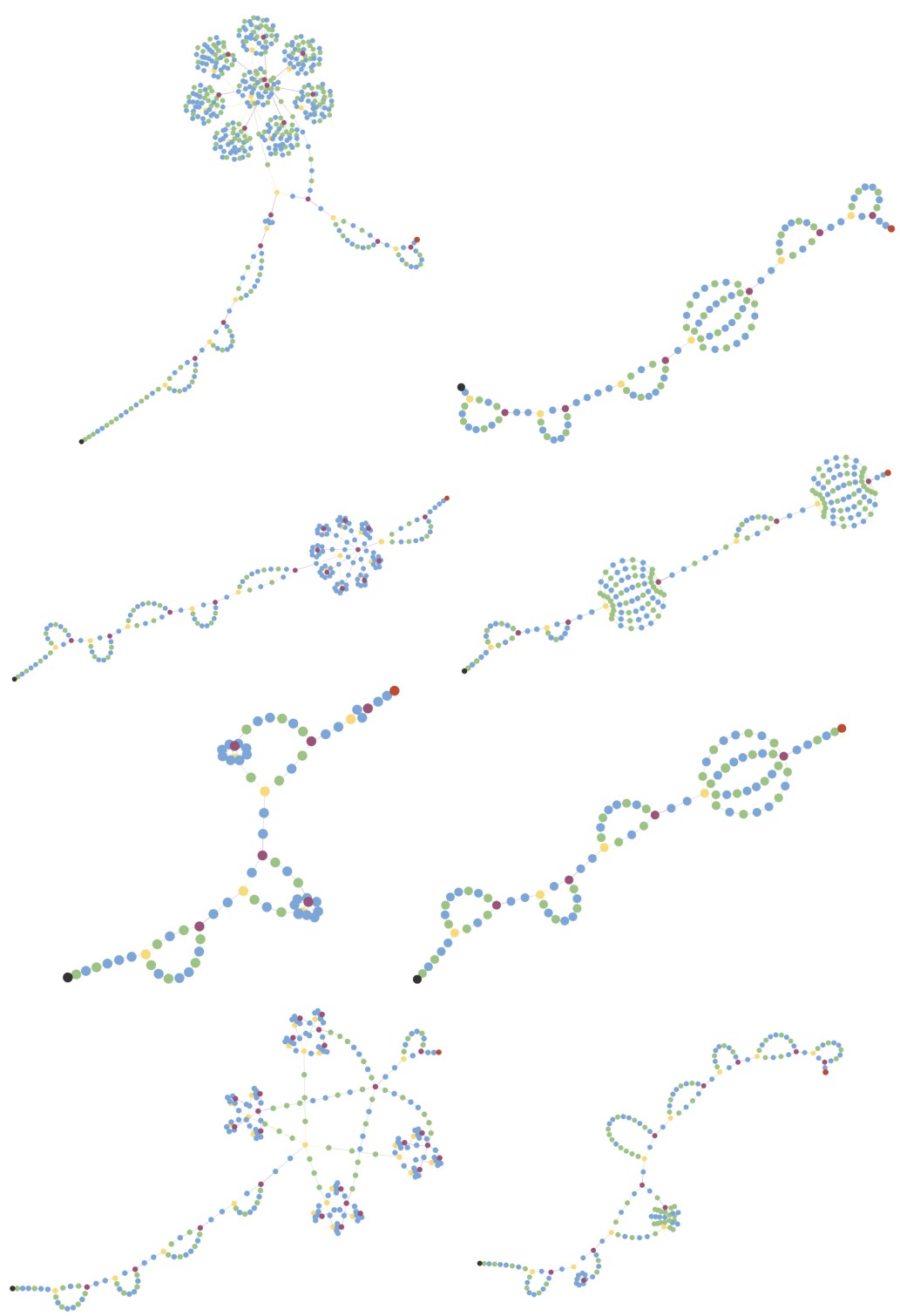

Figure 8: The top architectures found by RE(Mix) in `einspace` on Unseen NAS. From left to right, row by row: AddNIST, Language; MultNIST, CIFARTile; Gutenberg, Isabella; GeoClassing, Chesseract.

