# OpenReview forum: "einspace: Searching for Neural Architectures from Fundamental Operations"
_NeurIPS.cc/2024/Conference — NeurIPS 2024 poster_

### Official Review · Reviewer_c4rx · 2024-07-11

**Soundness:** 3
**Presentation:** 3
**Contribution:** 2
**Rating:** 4
**Confidence:** 4

**Summary:**

Neural Architecture Search (NAS) often produces incremental improvements due to limited diversity in traditional search spaces. To address this, the paper introduces "einspace," a versatile search space built from probabilistic context-free grammar (CFG). Einspace supports a wide range of architectures and operations, enabling the modelling of convolutions and attention mechanisms. Experiments on Unseen NAS datasets show that einspace can discover novel architectures from scratch and with grammars containing different foundation model architectures.

**Strengths:**

1. The paper is written very clearly and examples of CFGs aid understanding of the paper.
2. The authors release code which improves the reproducibility of the work and potential future work
3. While I do think that this paper opens up new opportunities and challenges for scaling NAS from scratch to larger spaces and datasets, I have questions (check questions) for the authors to improve the quality of their evaluations and make a stronger case for the practical applicability of einspace.

**Weaknesses:**

1. **Contribution**: Currently I fail to see the main contribution of the paper to the area to "searching architectures from scratch" [1]. I find the representation  properties of the search space to be same as [1] (except the use of probabilistic CFGs). Moreover, the search schemes proposed random and evolutionary search are not very sample efficient in such large spaces compared to [1], which uses bayesian optimization with graph kernels.
2. **Evaluation**: The paper does not compare to natural baselines like [1]. Moreover I find the evaluation to be quite limited in terms of the datasets evaluated and the base models finetuned/evaluated on.

[1] Schrodi, S., Stoll, D., Ru, B., Sukthanker, R.S., Brox, T. and Hutter, F., 2022. Towards discovering neural architectures from scratch.

**Questions:**

Questions:

1, The paper claims that it is the first one to represent foundation model architectures with a NAS search space. However [1] already represented a language model using CFGs. Also [1] can potentially present any architecture (ViT, MLPMixer). Is my understandng correct that this paper introduces a PCFG on [1] to  limit the search space size? I am also currently lacking a CFG which is a union of multiple foundation model architectures in a single grammar. Could the authors provide an example for this?

2. The paper uses random search and evo search, both of which are sample inefficient and do not scale well with the size of the dataset and search space. How do more sample efficient bayesian optimization algorithms perform here?

3. Currently the standard for small unseen datasets is finetuning a pretrained model on the smaller downstream task. Could the authors should ideally compare to for eg: simply finetuning a pretrained vision transformer or efficientnet on the unseen dataset?

4. Evaluation on larger datasets is missing. Architectures which are able to exploit dataset patterns change depending on the scale of datasets available (eg: Vision transformers [2]). Are the architectures discovered significantly different from the existing ones even when the dataset is scaled up. How efficient is the search on this scale?

5. Could you present parameter and FLOPs counts in Table 1?

6. How is the branching rate hyperparameter set for a newer search space?





[1] Schrodi, S., Stoll, D., Ru, B., Sukthanker, R.S., Brox, T. and Hutter, F., 2022. Towards discovering neural architectures from scratch.

[2] Raghu, M., Unterthiner, T., Kornblith, S., Zhang, C. and Dosovitskiy, A., 2021. Do vision transformers see like convolutional neural networks?. Advances in neural information processing systems, 34, pp.12116-12128.

**Limitations:**

I do think that the paper opens up newer avenues of research in neural architecture search, however, given the limited evaluation I do have concerns about the applicability of the method. Check weakness and questions for details on the limitations. I am happy to raise my score if each of my concerns are appropriately addressed.

---

> ### Author Rebuttal · Authors · 2024-08-07
>
> We thank the reviewer for their valuable feedback, and respond to each point below.
>
> _Contribution:_ There are three key differences between einspace and the CFG-based spaces of Schrodi et al [1].
> - Our space unifies multiple architectural families (ConvNets, Transformers, MLP-only) into one single expressive space while [1] present variations of their spaces centred around ConvNets only, with a separate instantiation focusing only on Transformers.
> - einspace extends to probabilistic CFGs. This constitutes a significant contribution by enabling a set of benefits that include (i) allowing experts to define priors on the search space via probabilities, and (ii) enabling a broader range of search algorithms that incorporate uncertainty estimates inside the search itself.
> - einspace contains recursive production rules (e.g. M -> MM), meaning the same rule can be expanded over and over again, providing a very high flexibility in macro structures. [1] instead focuses on fixed hierarchical levels that limits the macro structure to a predefined (though very large) set of choices.
> We will ensure that these differences are better highlighted in the manuscript.
>
> _Comparing to baseline search space:_ We appreciate the reviewer’s concern and provide a comparison with the baseline of [1]. Table A presents results comparing einspace to the CFG-based, hNASBench201 from Schrodi et al. This allows for a fairer and more direct comparison of the search spaces. These results show how einspace compares favourably under the same evolutionary search. Overall, we highlight that our search results on einspace are competitive, even with far weaker priors on the search space.
>
> Table A: Comparing einspace to hNASBench201 (from Schrodi et al. [1])
> ||RE(hNASBench201)|RE(Mix)(einspace)|
> |-|-|-|
> |AddNIST|93.82|97.72|
> |Language|92.43|97.92|
> |MultNIST|93.44|92.25|
> |CIFARTile|58.31|62.76|
>
> _Foundation model architectures:_ We apologise for any confusion caused, as we do not wish to claim that we are the first to represent foundation models within a NAS search space. We claim that we are the first to represent multiple architectural families (specifically ConvNets, Transformers, MLP-only) in a unified search space. The spaces presented in [1] focus on either ConvNets or Transformers, but not both in a unified space. If there is a particular wording that caused this confusion, we are happy to rephrase it to be more clear.
>
> _PCFG extension:_ See previous answer above.
>
> _Unified grammar:_ We apologise for any confusion here too. We only present one grammar in the paper, and it is shown in section 3.3. This grammar is expressive enough to be the union of multiple foundation model architectures in a single grammar. All experiments and all figures in the paper are from architectures generated by that grammar. Please let us know if any specific wording caused this confusion, and we will be happy to improve it.
>
> _Sample efficiency and BO:_ Due to the large set of experimental requests and queries, suggested by our six reviewers, we unfortunately did not have enough time nor spare compute to explore sample efficient BO during the time limited rebuttal period. We conjecture that such sample efficiency strategies may combine well with our large search space and provides for an exciting future direction.
>
> _Finetuning pretrained models:_ We thank the reviewer for their suggestion, and agree that this is an interesting comparison. Below we present results for the finetuning of the ResNet18 and EfficientNet-B0 architectures on the Unseen NAS datasets. The results show that we can often get a significant boost from finetuning, but for datasets that differ too much from the pretraining task (ImageNet) such as Language and Gutenberg, there is actually a degradation in performance. These are in fact the datasets where we see some of the biggest improvements from using einspace, and it highlights the expressiveness of our new search space.
>
> Table B: Finetuning pretrained models
>
> ||RN18|FT(RN18)|FT(EfficientNetB0)|
> |-|-|-|-|
> |AddNIST|93.36|94.69|94.77|
> |Language|92.16|90.31|90.62|
> |MultNIST|91.36|91.12|91.90|
> |CIFARTile|47.13|52.26|79.32|
> |Gutenberg|43.32|42.52|42.08|
> |Isabella|63.65|62.35|67.46|
> |GeoClassing|90.08|90.70|95.81|
> |Chesseract|59.35|61.29|62.24|
>
> _Larger datasets:_ In the rebuttal we present an array of additional experimental results. Our updated evaluation now covers 16 different datasets, with sizes ranging from thousands of data points to over a million ('Satellite', NB360), and spatial resolutions of up to 256x256 (‘Cosmic’, NB360). Unfortunately we did not have enough time nor spare compute to explore e.g. ImageNet, but we believe our evaluation is now broad enough to highlight the expressiveness and utility of einspace.
>
> _Parameter counts and FLOPs:_ Thank you for this suggestion, parameter counts are now presented in Table C.
>
> Table C: Parameter counts
>
> ||AddNIST|Language|MultNIST|CIFARTile|Gutenberg|Isabella|Geoclassing|Chesseract|
> |-|-|-|-|-|-|-|-|-|
> |DrNAS|4M|4M|5M|3M|3M|4M|4M|4M|
> |PC-DARTS|3M|-|3M|3M|2M|2M|-|2M|
> |RE(Mix)|20M|1M|25M|5M|1M|5M|4M|11M|
>
> _Setting branching rate hyperparameter:_ We assume the reviewer is asking “how is the branching rate hyperparameter set for a new task?”. In this case, we can confirm that the branching rate is set only once, based on the theoretical guidance of Section 3.7. We highlight that this hyperparameter remains constant across -all- of our experiments and tasks. We believe this evidences the generalisability of our search space, and its ease of use. If we have misunderstood the query, we would ask the reviewer to further clarify on this point. Thank you.
>
> We thank the reviewer for detailed comments that help us improve our manuscript. We have provided additional experiments that significantly extend our evaluation. We hope we have addressed all fundamental questions raised and, in light of our clarifications, we ask that the reviewer considers increasing their score.

---

> ### Comment · Reviewer_c4rx · 2024-08-12
> **Response to rebuttal**
>
> I appreciate the efforts of the authors in addressing my questions I am increasing my score to 4. That said, I am still concerned about the parameter sizes of architectures discovered by einspace in most cases being much larger than architectures being compared with.

---

> > ### Author Response · Authors · 2024-08-12
> > **Response to reviewer**
> >
> > We thank the reviewer for their response and updated scores. We appreciate that there may be concern that at times our method finds architectures of higher parameter counts. However, we argue that this shows the flexibility of the search space in adapting to the difficulty of the given task. On three datasets (AddNIST, MultNIST and Chesseract) the parameter count is significantly increased, but in other cases it is significantly reduced (Language and Gutenberg). The increases could be easily controlled by either rejecting architecture candidates that are too big, or by including an efficiency component to the optimisation objective.

---

### Official Review · Reviewer_ZVSn · 2024-07-16

**Soundness:** 3
**Presentation:** 3
**Contribution:** 3
**Rating:** 6
**Confidence:** 3

**Summary:**

This paper proposes a new type of neural architecture search (NAS) search space that goes beyond standard, small NAS search spaces. Most popular search spaces in NAS are quite small, and include architectures that include known motifs because they are built around standard architectures. The authors point out that this is why NAS has failed to produce fundamentally new architectures, and instead, that most advancements in neural network architectures are from expert driven hand-design. Their search space, einspace, is a probabilistic context-free grammar (CFG) of fundamental operations.

**Strengths:**

- The paper is well-written, well-motivated, and easy to follow.
- The paper is clear in that its innovation is related to search space design, with many suggestions for possible search algorithms. The search space itself is quite impressive, and includes both convolutions as well as self-attention operations (built-up from more fundamental building blocks).
- The results of search seem impressive -- while einspace does not always yield stronger architectures compared to other methods or hand-designed baselines, it does sometimes yield significantly better architectures. Furthermore, the comparison to the random search baseline is appreciated.
- The search space supports initialization using existing architectures -- this is important for setting priors for problems in which good architectures are already known.

**Weaknesses:**

- The resulting architectures in Table 1 do not always outperform SOTA architectures, but when they do, the improvement can be significant. On the other hand, there does not seem to be a comparison of the computational costs involved -- this would be helpful to include, especially as the search space is quite large.
- The authors include results on NAS-Bench-360, which is great, however the results are incomplete and it is unclear why the authors only evaluate on 5 out of the 10 tasks. This seems particularly important as the motivation of einspace is similar to the line of work related to NAS-Bench-360.
- In order to make search tractable, it seems like many assumptions are made on the priors on the search space. This makes sense, however, it would be interesting to explore variations on these choices more in-depth.

If the authors adequately address these concerns, I will gladly raise my score.

**Questions:**

- Out of curiosity, what do the authors view as the key reason for einspace not being able to express RNNs and SSMs? This is a valid design decision, but it also seems as though it should be fairly straightforward to design a version of einspace that supports recurrence (I could be mistaken about this).
- Why did the authors choose the 5 NAS-Bench-360 tasks that were used in their evaluation? Does the method work on the other tasks?

**Limitations:**

The authors clearly state the limitations, and even discuss avenues for future work. This includes extending the search space to handle recurrent architectures and state space models.

---

> ### Author Rebuttal · Authors · 2024-08-07
>
> We thank the reviewer for their valuable feedback, and respond to each point below.
>
> _Computational costs:_ We thank the reviewer for the useful suggestion. In the table below we include search time results for NAS methods DrNAS, PC-DARTS and RE(Mix), that were originally listed in Tab.1. Note that two numbers are missing due to missing logs from the authors of [2]. We see that, as expected, the gradient-based DrNAS and PC-DARTS are significantly faster compared to the black box RE(Mix) which trains 500 networks independently. We update our manuscript to report these results and add some further discussion on the tradeoff between diversity and time complexity in light of these observations, towards providing the reader with additional insight into the related issues.
>
> Table D: Time-consumption (in hours)
> | AddNIST | Language | MultNIST | CIFARTile | Gutenberg | Isabella | GeoClassing | Chesseract |
> |-|-|-|-|-|-|-|-|
> | DrNAS | 10 | 9 | 11 | 25 | 13 | 59 | 23 | 10 |
> | PC-DARTS | 4 | - | 5 | 12 | 9 | 30 | - | 2 |
> | RE(Mix) | 55 | 71 | 32 | 62 | 42 | 80 | 65 | 42 |
>
> _NAS-Bench-360 tasks missing:_ We appreciate the reviewer's concern regarding our original NAS-Bench-360 evaluation. Due to resource and time constraints we did not manage to have all NAS-Bench-360 results ready. The five tasks we reported in our original submission were the most amicable to our search space and required no adjustments.
>
> In this rebuttal, we present further results on 1D datasets within the benchmark, which required us to adjust the einspace CFG to make it compatible. Essentially, these adjustments include replacing our decomposed convolutional operators with 1-dimensional versions. We add the results for these 1D tasks, along with details of these minor adjustments, to our updated manuscript. The remaining three datasets, within the benchmark, are in progress and will also be included for the camera-ready submission.
>
> Table A: Additional datasets from NAS-Bench-360 (one-dimensional)
> | | WRN | DARTS (GAEA) | Expert | RE(WRN) einspace |
> |-|-|-|-|-|
> | Satellite | 15.29 | 12.51 | 19.80 | 12.55 |
> | DeepSea | 0.45 | 0.36 | 0.30 | 0.36 |
>
> In the rebuttal in general, we present an array of additional experimental results, towards strengthening our submission. Our updated experimental evaluation now covers 16 different datasets, with sizes ranging from thousands of data points to over a million ('Satellite', NB360), and spatial resolutions of up to 256x256 (‘Cosmic’, NB360). We provide further evidence of the efficacy of our proposed space, including more expensive tasks, and the experimental breadth can be considered of high diversity. We believe this serves to strengthen our motivation for einspace, that is indeed aligned with works related to NAS-Bench-360 and we thank the reviewer for the suggestion.
>
> _Exploring design priors of einspace:_ We agree with the reviewer that this provides an additional interesting direction for exploration. Towards exploring this aspect further, we present results comparing einspace to the previous, CFG-based, hNASBench201 from Schrodi et al. This allows for an initial study on the effects of our search space design choices and, in particular, the increased expressiveness compared to hNASBench201. These results show how einspace compares favourably to a different search space under the same evolutionary search. Overall, we highlight that our search results on einspace are competitive, even with far weaker priors on the search space.
>
> Table A: Comparing einspace to hNASBench201 (from Schrodi et al. [1])
> || RE (hNASBench201) | RE (Mix) (einspace) |
> |-|-|-|
> | AddNIST | 93.82 | 97.72 |
> | Language | 92.43 | 97.92 |
> | MultNIST | 93.44 | 92.25 |
> | CIFARTile | 58.31 | 62.76 |
>
> We update our manuscript with these findings with the aim to further improve reader understanding for choices relating to search space priors and thank the reader for the suggestion.
>
> Questions:
>
> _Recurrence missing:_ The reviewer raises an interesting question and we agree that recurrence, and related operations, make for an appealing einspace extension. We believe recurrent operations can likely be integrated via the inclusion of a recurrent module that repeats the computation of the components within it; however we leave more detailed exploration of this direction to future work.
>
> _Why not full NAS-Bench-360?:_ As already mentioned, the reasons for only including 5 out of 10 NAS_Bench-360 tasks was due to limited time and compute resources, and that some of the tasks require further adjustments to our CFG codebase. We evidence here that we have been able to successfully extend our experimental work on this axis and further, that the full set of tasks will be included in any camera-ready version.
>
> We thank the reviewer for detailed comments that (1) thoroughly test the experimental and theoretical underpinnings of our ideas and (2) enable us to update our manuscript towards further improving clarity.
>
> We hope we have addressed all fundamental questions raised and, in light of our clarifications, we invite the reviewer to consider increasing their score.
>
>
> [1] Construction of Hierarchical Neural Architecture Search Spaces based on Context-free Grammars, Schrodi et al, NeurIPS 2023.

---

> > ### Author Response · Authors · 2024-08-13
> > **Message to reviewer ZVSn**
> >
> > Thank you again for taking the time to review our paper. We hope our response has addressed your concerns, and would be very grateful if you would reconsider your scores. If there are further questions we are happy to continue discussing until the deadline tomorrow.

---

### Official Review · Reviewer_wLBZ · 2024-07-17

**Soundness:** 4
**Presentation:** 3
**Contribution:** 3
**Rating:** 6
**Confidence:** 4

**Summary:**

This paper proposes a new search space named einspace based on a parameterized probabilistic context-free grammar for neural architecture search. In contrast to conventional search space composed of high-level operations and modules such as recurrent layer, convolutional layer, and activation layer, einspace consists of more fundamental operations, such as clone, summation, permutation, and activation. This einspace can be defined by a context-free grammar and can represent the neural architecture as the derivation tree. The experiments on Unseen NAS and NAS-Bench-360 demonstrate the advantage of this search space.

**Strengths:**

1. The paper is well-written and is easy to follow.
2. The idea of fundamental operations and context-free grammar for search space is great and interesting.
3. This search space has the advantage of designing new architectures based on existing SOTA architectures such as ResNet, Wide-ResNet, ViT, and MLP-Mixer.

**Weaknesses:**

Despite the strengths of this paper, I still have some concerns that prevent me from confidently recommending acceptance:
1. The first problem is about the novelty. Since einspace is not the first search space based on CFG, and the fundamental operations of einspace are common in the existing search space, the novelty of einspace may not reach my expectations. Although the fundamental operations are different from and more expressive than the high-level operations and rigid structures of existing search space, these fundamental operations are also commonly used in existing search space. Furthermore, it is complicated to represent convolution and skip connection using these operations and it is impossible to represent recurrent computation, as the authors discussed in the Limitation section.
2. The second problem is about the tradeoff between diversity and time complexity of the search space. It would be better to analyze the time complexity or compare the time consumption between einspace and others, such as the space of DARTS.
3. From the experimental results, the architectures searched in einspace do not seem to be that competitive compared to other methods, as shown in Table 1. I think the results are also influenced by the search strategy. Is it possible to apply RL or differentiable search strategies to this search space? If possible, the author can give a table listing which search strategies are suitable for einspace and which search strategies are better.

**Questions:**

See weaknesses.

**Limitations:**

The authors have discussed the limitations in their paper. Besides, there are no potential negative societal impacts of this work.

---

> ### Author Rebuttal · Authors · 2024-08-07
>
> We thank the reviewer for their valuable feedback, and respond to each point below.
>
> _Novelty:_ We acknowledge previous CFG-based studies exist, however we would highlight that we are the first work to introduce the concept of pCFGs for search space design, which we show brings advantages in controlling the expected architecture complexities. We are also the first to unify multiple architectural families into one single expressive space. We provide direct experimental evidence, namely that regularised evolution seeded with diverse SOTA architectures performs competitively across a broad range of datasets. We believe that this constitutes a meaningful step forward, beyond previous work on CFG based search space design.
>
> The fundamental operations in einspace are atomic, and rather than focussing on the fact that they are pre-existing, we highlight that the novelty of our work stems from our thinking around the required expressiveness of the search space. The unique manner in which we organise and leverage such atomic operators results in the emergence of new search spaces that contribute a previously unseen expressiveness for NAS-based tasks.
>
> _Complexity of components:_ We agree with the reviewer that existing hand-designed high-level operators (e.g. convolutions, skips) often require a complicated composition of atomic operators and that this typically necessitates specific human expertise. This observation is actually at the very crux of our argument. Rather than assuming that such coarse operators are optimal, and hiding their inner workings from the model, we instead enable the autonomous construction and discovery of such related operators.
>
> _Recurrence missing:_ While we account for a large set of important architectures, we are also clear to communicate that this is a non-exhaustive set. We do not view this as a large limitation, rather, it opens the door for interesting follow-up work, such as support for recurrent operations (e.g. via inclusion of a recurrent module that repeats the computation of the components within). We leave such directions for future work.
>
>
> _Tradeoff between diversity and time complexity:_ We thank the reviewer for the useful suggestion. In the table below we include search time results for NAS methods DrNAS, PC-DARTS and RE(Mix), that were originally listed in Tab.1. Note that two numbers are missing due to missing logs from the authors of [2]. We see that, as expected, the gradient-based DrNAS and PC-DARTS are significantly faster compared to the black box RE(Mix) which trains 500 networks independently. We update our manuscript to report these results and add some further discussion on the tradeoff between diversity and time complexity in light of these observations, towards providing the reader with additional insight into the related issues.
>
> Table D: Time-consumption (in hours)
> | AddNIST | Language | MultNIST | CIFARTile | Gutenberg | Isabella | GeoClassing | Chesseract |
> |-|-|-|-|-|-|-|-|
> | DrNAS | 10 | 9 | 11 | 25 | 13 | 59 | 23 | 10 |
> | PC-DARTS | 4 | - | 5 | 12 | 9 | 30 | - | 2 |
> | RE(Mix) | 55 | 71 | 32 | 62 | 42 | 80 | 65 | 42 |
>
> _Competitiveness:_ We highlight that, in Table 1, RE(RN18) and RE(Mix) achieve average ranks that are only beaten by BonsaiNet, which shows the high performance of our approach. Additionally, as part of our new results, we also present a direct comparison to hNASBench201, the hierarchical CFG-based search space from Schrodi et al [1]. These results show how einspace compares favourably to a different search space under the same evolutionary search. Overall, we highlight that our search results on einspace are competitive, even with far weaker priors on the search space.
>
> Table A: Comparing einspace to hNASBench201 (from Schrodi et al. [1])
> || RE (hNASBench201) | RE (Mix) (einspace) |
> |-|-|-|
> | AddNIST | 93.82 | 97.72 |
> | Language | 92.43 | 97.92 |
> | MultNIST | 93.44 | 92.25 |
> | CIFARTile | 58.31 | 62.76 |
>
> _Potential search strategies:_ We thank the reviewer for the idea of a table clarifying the potential search strategies for einspace. Like hNASBench201 from Schrodi et al [1], einspace is too large for one-shot methods that require all architectures to be instantiated into a single supernet. However, there are other weight-sharing methods relating to MCTS and SPOS that may be applicable.
>
> Table A: Comparison of einspace with existing search spaces. † Gradient-based search is difficult in these spaces due to their size, but other weight-sharing methods may be available. *The paper introducing hNASBench201 [1] also considers versions of the search space for Transformer language models.
>
> | | Type | Size | Focus | RS | RE | RL | BO | Gradient-based |
> |-|-|-|-|-|-|-|-|-|
> | einspace | pCFG | Huge | ConvNets, Transformers, MLP-only | ✓ | ✓ | ✓ | ✓ | † |
> | hNASBench201 | CFG | 10^446 | ConvNets* | ✓ | ✓ | ✓ | ✓ | † |
> | NASBench201 | Cell | 10^4 | ConvNets | ✓ | ✓ | ✓ | ✓ | ✓ |
> | NASBench101 | Cell | 10^5 | ConvNets | ✓ | ✓ | ✓ | ✓ | ✓ |
> | DARTS | Cell | 10^18 | ConvNets | ✓ | ✓ | ✓ | ✓ | ✓ |
>
> In light of our responses and improved evaluation, we invite the reviewer to consider increasing their score.
>
> [1] Construction of Hierarchical Neural Architecture Search Spaces based on Context-free Grammars, Schrodi et al, NeurIPS 2023.
> [2] Insights from the Use of Previously Unseen Neural Architecture Search Datasets, Geada et al, CVPR 2024.

---

> > ### Comment · Reviewer_wLBZ · 2024-08-12
> > **Response to Authors**
> >
> > Thanks to the authors for the detailed rebuttal. Although there are some issues of einspace, such as high time complexity and complex representation of skip connections, I think it is a valuable exploration of the search space with atomic operations. Therefore, I would like to raise my score to Weak Accept.

---

> > > ### Author Response · Authors · 2024-08-12
> > > **Response to Reviewer wLBZ**
> > >
> > > We thank the reviewer for their response and updated scores. If there are any outstanding concerns regarding the time complexity or representations of architectural components, we are happy to discuss further during the discussion period.

---

### Official Review · Reviewer_sBZH · 2024-07-17

**Soundness:** 2
**Presentation:** 2
**Contribution:** 2
**Rating:** 4
**Confidence:** 3

**Summary:**

The manuscript presents "einspace," a novel neural architecture search (NAS) space based on a parameterized probabilistic context-free grammar (CFG). The authors aim to address the limitations of current NAS methods by proposing a highly expressive search space that supports diverse network operations and architectures of varying sizes and complexities.

**Strengths:**

1. The paper introduces a unique NAS search space, "einspace," which is designed to be highly expressive, accommodating a wide range of architectures, including those not traditionally found in NAS literature.

2. The work contributes to the ongoing discussion on the role of search space expressivity and strategic search initialization in NAS, potentially paving the way for future research in this direction.

**Weaknesses:**

1. While the authors claim to introduce a new search space, the manuscript's Method Section appears to describe a set of rules that break down operators into smaller elements, which could be misinterpreted as a mere decomposition rather than a novel search space construct. This raises questions about the actual size and scope of the proposed search space, which would benefit from further clarification.

2. Section 3.7 is not clearly articulated, and the authors are encouraged to provide a simplified explanation of its main content to aid reader comprehension.

3. The primary goal of the proposed method is to design a highly expressive yet constrained search space. To substantiate this claim, it would be beneficial to conduct searches and validating on larger-scale datasets, such as ImageNet, which could more effectively demonstrate the superiority of the proposed search space. The current experiments on smaller datasets may not fully showcase the advantages of the search space.

4. The experimental settings are somewhat unclear. It appears that the authors search within the proposed space and then validate the discovered network structures on other datasets, such as those from NAS-Bench-360. However, NAS-Bench-360 imposes constraints on the design space that may not be compatible with the structures proposed by the authors. Further clarification on how the proposed space's network structures are adapted or validated on NAS-Bench-360 is needed to ensure the experiments are methodologically sound.

**Questions:**

NA

---

> ### Author Rebuttal · Authors · 2024-08-07
>
> We thank the reviewer for their valuable feedback, and respond to each point below.
>
> _Size and scope of the search space:_ We thank the reviewer for the comment however believe this point to be largely a matter of semantics. By decomposing coarse grained building blocks into atomic operators we meaningfully increase the size, complexity and flexibility of the search space. We evidence that this allows points in our search space to embody (common) architectures that cannot be represented in previously explored spaces. We note that multiple co-reviewers are impressed by the size and scope of our proposed space (ZVSn, wLBZ) and that our experiments serve to evidence the complexity of the search space (duvy).
>
> _Clarity of Section 3.7:_ We thank the reviewer for the opportunity to provide a simplified explanation. If we consider network architectural design to be a procedural, decision-making process, then the crux of the message in Sec. 3.7 is that: **we can introduce a probabilistic choice at each step in this process, to help us achieve the desirable level of architecture complexity**. At each step, there is a chance to continue building the network (adding more components) or to stop and finalise a part of it. The probability P(M -> C|M) is particularly important for this. Further, guided by previous work on PCFGs, we can carefully adjust these probabilities to provably control the average complexity of the generated architectures. In essence, the probabilistic approach strikes a balance between creating deep, complex networks and shallow, simpler ones. This helps to explore a wider range of architectural possibilities while maintaining control over the overall complexity of the generated models. We refine our phrasing of Sec. 3.7, towards further aiding reader understanding on this point.
>
> _Larger-scale datasets:_ We appreciate the reviewer's concern regarding limited evaluation and would firstly note that this is a common problem of previous NAS works, which often only consider a small number of datasets e.g. CIFAR10. To alleviate this concern we present an array of additional experimental results in our rebuttal, towards strengthening our submission. Our updated experimental evaluation now covers 16 different datasets, with sizes ranging from thousands of data points to over a million ('Satellite', NB360). Our updated experimental work provides further evidence of the efficacy of our proposed space, including more expensive tasks, and the experimental breadth can be considered more diverse than most previous NAS work that we are aware of. We address the issue regarding development of more efficient search strategies above (see reply to **duvy**) and accordingly defer evaluation of ImageNet scale tasks to future work.
>
> _Unclear experimental settings:_ We think there may be a misunderstanding on how we performed our experiments on NAS-Bench-360 for Table 5. These results are run independently to those in Table 1, and there is no transfer or adaptation between tasks. Throughout our evaluation, every search is performed on the same dataset that the evaluation is performed on. Thank you for highlighting this, we will revise the manuscript to make it clearer.
>
> Some datasets in NAS-Bench-360 do indeed impose some additional constraints on the search. The 5 tasks we considered for the submitted version were the easiest to use and required no adjustments to the search space. The Cosmic and Darcy Flow datasets simply needed a dense prediction output layer instead of a classifier.
>
> In our updated results in this rebuttal, we also consider some 1D datasets within the benchmark, and this required us to adjust the einspace CFG to make it compatible. Primarily, these adjustments include replacing our decomposed convolutional operators with 1-dimensional versions. The details of all adjustments will of course be added to the manuscript along with the results on these 1D tasks.
>
> Table A: Additional datasets from NAS-Bench-360 (one-dimensional)
> | | WRN | DARTS (GAEA) | Expert | RE(WRN) einspace |
> |-|-|-|-|-|
> | Satellite | 15.29 | 12.51 | 19.80 | 12.55 |
> | DeepSea | 0.45 | 0.36 | 0.30 | 0.36 |
>
> We thank the reviewer for the detailed comments and suggestions that enable us to update our manuscript towards further improving clarity. We hope we have addressed all fundamental questions raised and, in light of our clarifications, we invite the reviewer to consider increasing their score.

---

> > ### Comment · Reviewer_sBZH · 2024-08-11
> > **Thanks for the rebuttal.**
> >
> > After reading the rebuttal, I raise my score to 4 and lower my confidence. There is something I have misunderstood but I fail to figure it out.

---

> > > ### Author Response · Authors · 2024-08-12
> > > **Thanks for the response**
> > >
> > > We thank the reviewer for their response and for updating their scores. Let us know if there is anything we can clarify further to clear up the misconceptions, and we'll be happy to do so in the remaining discussion period.

---

### Official Review · Reviewer_s1qw · 2024-07-19

**Soundness:** 3
**Presentation:** 3
**Contribution:** 2
**Rating:** 5
**Confidence:** 4

**Summary:**

This paper introduces einspace, a hierarchical space of neural architectures based on parametric probabilistic context free grammar, which is expressive enough to accommodate various state-of-the-art architectures including ResNets and Transformers.

The Authors further perform Regularized Evolution (RE) search over einspace either searching from scratch or seeding the initial population with state-of-the-art architectures. In particular when seeded with ResNet18, RE generates novel architectures that significantly outperform ResNet18 on multiple datasets from Unseen NAS. Furthermore, RE both seeded with ResNet18 and Mix (a mixture of SOTA architectures) significantly outperform RE from scratch as well as random sampling and random search.

**Strengths:**

This study extends previous research on hierarchical architecture spaces by introducing a more expressive framework capable of accommodating diverse structures including convolutional networks and transformers.

Novelties include imposing minimal priors which facilitate successful search within a highly expressive architecture space. Additionally, grammar rules are equipped with parameters to ensure the generation of valid architectures through the combination of various components. Moreover, the complexity / depth of architectures are regulated by tuning the probabilities assigned to production rules.

The authors demonstrate through a number of experiments involving ResNet and WideResNet architectures that network performance can be significantly enhanced by utilizing einspace in conjunction with RE, when seeding the initial population with these SOTA architectures.

**Weaknesses:**

The experiments are currently limited. While the paper does not introduce a search strategy tailored to this search space, it is crucial to emphasize experiments demonstrating the potential for RE, possibly seeded with SOTA networks, to enhance model performance.

The current evaluations focus on ResNet18 on the Unseen NAS datasets as well as WRN on datasets from NASBench360. Extending this analysis to include a broader range of models, e.g. those listed in Table.1 of Unseen NAS, such as AlexNet and DenseNet, would provide useful insights on the effectiveness of einspace.

Moreover, it would be valuable to explore the application of RE on einspace to improve performance on widely used datasets like CIFAR10, e.g with ResNet and in particular ViT,  given that the paper highlights the capability of einspace to support transformer architectures as an advantage.

**Questions:**

How does einspace compare with other hierarchical search spaces in the literature [1] and [2] in terms of performance? To support the advantages of einspace, including expressivity at a reasonable search cost, it would be beneficial to conduct comparative evaluations, at least with RE, across a number of tasks.

Does it make sense to apply black-box search methods other than RS, RE to einspace, for example the bayesian optimization-based methods BOHNAS and NASBOWL used in [2]?

Approximate search times for RE on datasets of Table.1 are reported in appendix B.3. How do these search times compare with those of other NAS methods listed in Table.1? A similar comparison would also be valuable for the experiments in Table 5.

[1] Hierarchical Representations For Efficient Architecture Search (Simonyan et.al 2018)
[2] Construction of Hierarchical Neural Architecture Search Spaces based on Context-free Grammars (Schrodi et.al 2023)

**Limitations:**

The limitations are adequately discussed in the final section.

---

> ### Author Rebuttal · Authors · 2024-08-07
>
> We thank the reviewer for their valuable feedback, and respond to each point below.
>
> _Limited evaluation:_ We present additional experimental evaluation in this rebuttal on multiple axes. Taking into account all new results, our experimental evaluation now covers 16 different datasets, with sizes ranging from thousands of data points, to a million (Satellite from NB360), and spatial resolutions of up to 256x256 (‘Cosmic’, NB360). The new results further evidence the efficacy of einspace and our seeded RE and we believe our study is now significantly broader and more diverse than most NAS work we are aware of. We will update our manuscript to include the additional experimental work and thank the reviewer for the suggestion.
>
> _Broader range of models:_ Our rebuttal now provides initial results on RE(DenseNet121) using einspace. We see gains here as well, especially on the Language dataset, where it almost matches the performance of RE(RN18)=96.84. We believe the further exploration of additional models constitutes potentially valuable future work.
>
> Table A: DenseNet121 results
> | | DenseNet121 | RE(DenseNet121) |
> |-|-|-|
> | AddNIST | 94.72 | 94.84 |
> | Language | 91.26 | 96.42 |
>
> _More datasets eg. CIFAR10 with ResNet and ViT:_ Our updated experimental evaluation now covers 16 different datasets, with sizes ranging from thousands of data points to over a million ('Satellite', NB360). Our updated experimental work includes more expensive tasks and provides further evidence of the efficacy of our proposed space. The experimental breadth has been meaningfully increased and can be considered to constitute a diverse range of tasks.
>
> We now present results on CIFAR10, using our regularised evolution seeded with Resnet18, and the mix of architectures (including a ViT). We can see that the improvement in this case is not as significant as for other datasets, an effect we attribute to the broad focus of einspace that goes beyond ConvNets, which have long been optimised for datasets like CIFAR10.
>
> Table B: CIFAR10 results
> | | RN18 | RE(RN18) | RE(Mix) |
> |-|-|-|-|
> | CIFAR10 | 94.91 | 95.31 | 94.73 |
>
> _einspace vs other hierarchical spaces:_ We agree that a direct comparison between einspace and previous CFG-based spaces would be beneficial. We are now happy to report new results comparing RE on einspace vs. RE on hNASBench201 (hierarchical+non-linear) from Schrodi et al [1] in the table below. The results show that our searches in einspace tend to outperform those on hNASBench201, and that both improve upon the baseline network. We thank the reviewer for this suggestion. The full set of results will be included in the updated manuscript.
>
> Table C: Comparing einspace to hNASBench201 (from Schrodi et al. [1])
> || RN18 | RE (hNB201) | RE (Mix) (einspace) |
> |-|-|-|-|
> | AddNIST | 93.36 | 93.82 | 97.72 |
> | Language | 92.16 | 92.43 | 97.92 |
> | MultNIST | 91.36 | 93.44 | 92.25 |
> | CIFARTile | 47.13 | 58.31 | 62.76 |
>
> _BO in einspace:_ Bayesian optimisation methods are certainly applicable to einspace. Similar to how BOHNAS is used in [1], we think a hierarchical kernel can work well with our CFG formulation. Unfortunately, due to resource and time constraints, we were not able to perform this experiment in time for this rebuttal. We conjecture that BO could provide a sample efficient search and be further improved through seeding with SOTA architectures, like we do with RE. We leave more in depth exploration of advanced search strategies to future work.
>
> _Search times:_ In the table below we include search time results for NAS methods DrNAS, PC-DARTS and RE(Mix), that were originally listed in Tab.1. Note that two numbers are missing due to missing logs from the authors of [2]. We see that, as expected, the gradient-based DrNAS and PC-DARTS are significantly faster compared to the black box RE(Mix) which trains 500 networks independently. We update our manuscript to report these results and add some further discussion on the tradeoff between diversity and time complexity in light of these observations, towards providing the reader with additional insight into the related issues. We thank the reviewer for the useful question.
>
> [1] Construction of Hierarchical Neural Architecture Search Spaces based on Context-free Grammars, Schrodi et al, NeurIPS 2023.
> [2] Insights from the Use of Previously Unseen Neural Architecture Search Datasets, Geada et al, CVPR 2024.
>
> Table D: Time-consumption (in hours)
> | AddNIST | Language | MultNIST | CIFARTile | Gutenberg | Isabella | GeoClassing | Chesseract |
> |-|-|-|-|-|-|-|-|
> | DrNAS | 10 | 9 | 11 | 25 | 13 | 59 | 23 | 10 |
> | PC-DARTS | 4 | - | 5 | 12 | 9 | 30 | - | 2 |
> | RE(Mix) | 55 | 71 | 32 | 62 | 42 | 80 | 65 | 42 |

---

> > ### Comment · Reviewer_s1qw · 2024-08-12
> >
> > I thank the Authors for their detailed response. Based on the additional results I would like to raise my score to 5.

---

> > > ### Author Response · Authors · 2024-08-12
> > > **Response to Reviewer s1qw**
> > >
> > > We thank the reviewer for their response and updated scores. If there are any outstanding concerns, we are happy to clarify further during the discussion period.

---

### Official Review · Reviewer_duvy · 2024-07-28

**Soundness:** 2
**Presentation:** 4
**Contribution:** 3
**Rating:** 6
**Confidence:** 5

**Summary:**

This paper introduces einspace, a search space that is designed to hierarchically encode architectures using probabilistic context-free grammars (PCFG). It can encode various architectural components, such as convolutions, attention mechanisms, etc. The authors demonstrate the efficacy of simple blackbox optimizers in einspace to discover architectures that perform competitively on various tasks and datasets.

**Strengths:**

In general the motivation to move beyond the conventional NAS spaces is valid and really important in my opinion. The paper is also very well-written, with simple examples followed by a more generic definition of the search space and interesting application to various tasks and datasets.
Discovering novel architectures is a very challenging problem for the NAS community and as far as I know, this has not been achieved yet. Having a complex and versatile search space is the first step towards this goal.

Some other positive aspects of this submission:

- Interesting experiments showing the complexity of the search space and the need of guided search methods (e.g. evolutionary strategies as the authors show), instead of random search, which on previous spaces (e.g. the DARTS one) was shown to perform already well.
- Available code that fosters reproducibility and enables easier future research on this topic.
- Extending the prior work of [1] to probabilistic CFG. This enables (1) easier expert prior definitions on the search space, (2) a broader range of algorithms applied on einspace, that can as well incorporate uncertainty estimates of their choices inside the search itself.

**Weaknesses:**

Despite the vast number of architectures that einspace includes, the major problem is how to search on these spaces in an efficient way. The prior work of [1] used BO with a hierarchical graph kernel (a novelty aspect of that paper), however that was still expensive, especially when moving to image classification tasks. I think defining search spaces that are very complex is very useful task, however, the NAS problem is not only solved by defining the search space alone. The search algorithm is a major component of the whole pipeline, and using blackbox methods to search in such search spaces will be computationally demanding and will still require company scale computation.

Below I list my major concerns about this submission:

- *Novelty*: I think the paper has some novel aspects compared to [1], e.g. the fact that can encode both attention-mechanisms and convolution, or other operators, or the probabilistic extension of the CFG. However, as the authors mention in the limitation section, it would be great if accompanying the search space, there would be a new proposed search method that can efficiently search on this space by exploiting the PCFG.
- *Limitations of the search space*: As the authors mention, there are various important architectures that cannot be encoded in einspace.
- *Claims*: I think the claim (for instance in the abstract) that you find "novel architectures" can mean many things. Correct me if I am wrong, but in my opinion einspace still encodes most of the known architectures, and searching on it will only re-discover them or improve on top of those architectures, but it won't find a completely novel architectural component, as for instance ResNets introduced the residual connection back then.
- *Limited experimental evaluation and results*: The empirical evaluations are not enough in my opinion. The used tasks are simple and one function evaluation is cheap enough to allow blackbox methods to run there. Evaluating on more expensive tasks would require the development of more efficient search strategies that are tailored to einspace. Moreover the results shown in Table 1 are not that impressive considering RE is typically a very strong baseline, so I would have expected it to outperform all other methods.

**Questions:**

- Could you please discuss further in more details on what are the main advantages of using CFG instead of graph-based or other encodings used in NAS [2]?


-- References --

[1] https://proceedings.neurips.cc/paper_files/paper/2023/file/4869f3f967dfe954439408dd92c50ee1-Paper-Conference.pdf

[2] https://proceedings.neurips.cc/paper/2020/file/ea4eb49329550caaa1d2044105223721-Paper.pdf

**Limitations:**

The authors have addressed the limitations adequately.

---

> ### Author Rebuttal · Authors · 2024-08-07
>
> We thank the reviewer for their valuable feedback, and respond to each point below.
>
> _Novelty:_ We agree that our core contributions relate to the search space. However we respectfully argue that the introduction of a valuable new space forms a meaningful and valid contribution, independently of presenting a completely novel search strategy. We provide an opportunity for the field to develop new search strategies for our space, and encourage interesting new lines of work. While we acknowledge the impressive work of Schrodi et al. [1], who offer both a new search space framework and search strategy, there are also several recent papers whose main contributions consist solely of a search strategy [2, 3], and this is naturally complemented by papers that contain a search space core contribution. As reviewer **duvy** also notes; our search space highlights the importance of search strategy choice, in relation to space complexity. We believe this further helps to encourage follow-up work on search strategies and we will explicitly strengthen this idea in our revision. Finally, our secondary contribution in this paper is that seeding the search with existing SOTA architectures is a powerful approach that has been previously overlooked. Our expressive search space makes this straightforward as it contains such a diverse set of existing architectures.
>
> [1] Construction of Hierarchical Neural Architecture Search Spaces based on Context-free Grammars, Schrodi et al, NeurIPS 2023.
> [2] ZARTS: On Zero-order Optimization for Neural Architecture Search, Wang et al, NeurIPS 2022.
> [3] PASHA: Efficient HPO and NAS with Progressive Resource Allocation, Bohdal et al, ICLR 2023.
>
> _Limitations of the search space:_ To date, einspace constitutes one of the most expressive search spaces in the NAS field. We evidence that it combines multiple powerful architectural families in a unified space; including ConvNets, transformers and MLP-only architectures. While we account for a large set of important architectures, we are also clear to communicate that this is a non-exhaustive set. We do not view this as a large limitation, rather, it opens the door for interesting follow-up work, such as support for recurrent operations (e.g. via inclusion of a recurrent module that repeats the computation of the components within). We leave such directions for future work.
>
> _Claims on novel architectures:_ We clarify that einspace has an 'increased ability to find novel architectures'. It includes both examples of existing SOTA architectures, like ResNets and ViTs, but importantly it includes a huge amount of architectures anywhere between and around these existing models. As an example, many previous search spaces consider the self-attention module to be a fixed component, while we model the intricacies of each matrix multiplication, activation, linear layer and branching/merging structures. By changing the number of branches, or the operations done in each branch, we are able to build and discover uncommon architectural components that are on a similar order of complexity as existing self-attention or indeed residual connections. That is; we provide a significantly more granular space within which novel components can be discovered. It is of course true that we impose some constraints on the space, as discussed in Sec. 3.4, but these are relatively weak constraints that don’t significantly reduce expressiveness. We are open to refining the specific phrasing of claims on this point, towards aiding understanding, if the reviewer deems this important.
>
> _Limited evaluation and results:_ We appreciate the reviewer's concern regarding limited evaluation and would firstly note that this is a common problem of previous NAS works, which often only consider a small number of datasets e.g. CIFAR10. To alleviate this concern we present an array of additional experimental results in our rebuttal, towards strengthening our submission. Our updated experimental evaluation now covers 16 different datasets, with sizes ranging from thousands of data points to over a million ('Satellite', NB360), and spatial resolutions of up to 256x256 (‘Cosmic’, NB360). Our updated experimental work provides further evidence of the efficacy of our proposed space, including more expensive tasks, and the experimental breadth can be considered more diverse than most previous NAS work. We address the issue regarding development of more efficient search strategies above (see previous point) and accordingly defer evaluation of ImageNet scale tasks to future work.
> As part of our new results, we also present a direct comparison to hNASBench201,  the hierarchical CFG-based search space from Schrodi et al [1]. These results show how einspace compares favourably to a different search space under the same evolutionary search. Overall, we highlight that our search results on einspace are competitive, even with far weaker priors on the search space.
>
> Table A: Comparing einspace to hNASBench201 (from Schrodi et al [1])
> || RE (hNASBench201) | RE (Mix) (einspace) |
> |-|-|-|
> | AddNIST | 93.82 | 97.72 |
> | Language | 92.43 | 97.92 |
> | MultNIST | 93.44 | 92.25 |
> | CIFARTile | 58.31 | 62.76 |
>
> _Encodings:_ Our CFG formulation encodes architectures in the form of derivation trees. This explicitly differs from a graph encoding; a derivation tree records the set of design choices that define a flexible macro structure for the architecture, while a graph encoding alternatively provides only a rigid macro-structure-blueprint for representable architectures (e.g. via fixed size adjacency matrices). Through the use of derivation trees, einspace allows for mutations that can effectively alter both the macro structure -and- the individual components of an architecture. Modifications of this class are more difficult if using rigid graph encodings. We will update our manuscript to include further discussion on this point and thank the reviewer for the helpful question.

---

> > ### Comment · Reviewer_duvy · 2024-08-12
> >
> > I thank the authors for their response and the additional experiments comparing einspace to the hierarchical NB201 space from Schrodi et al. I will increase my score, however, I think this paper has the potential to become a really strong publication by incorporating the feedback from the reviewers, and one more iteration might be beneficial in the long run.

---

> > > ### Author Response · Authors · 2024-08-13
> > > **Response to Reviewer duvy**
> > >
> > > We thank the reviewer for their response and their updated scores, and we are encouraged that they think our work has strong potential. We will integrate all feedback from this review process into our paper and for a potential camera-ready version we are also working towards evaluating a BO search strategy on our search space. Thank you for a great discussion.

---

### Author Rebuttal · Authors · 2024-08-07

We thank the six (!) reviewers for their time and valuable comments that improve the quality of our work. We are encouraged by the positive feedback, namely:

- Multiple reviewers appreciate the novelty of our core idea, to move beyond conventional NAS spaces (duvy, wLBZ, ZVSn)
- That our experimental results evidence our method efficacy (duvy, s1qw, ZVSn)
- Our work opens up new opportunities and future NAS research directions (sBZH, c4rx)
- A majority of reviewers note that the paper is well-written, well-motivated, and easy to follow (duvy, wLBZ, ZVSn, c4rx)
- Our use of helpful examples and clarity of explanation (duvy, c4rx)
- Multiple reviewers appreciate that we release our source code preemptively during the review period (duvy, c4rx)

We address individual reviewers’ concerns below inline and also offer a communal reply here in order to address several common and important points.

**Novelty**: Our core contribution is a novel search space, which we argue is a meaningful and valid contribution, independently of presenting a completely novel search strategy. This provides an opportunity for the community to develop new search strategies for our space and encourages interesting new lines of work. That said, as a starting point, we have highlighted the effectiveness of seeding search with SOTA architectures, which is straightforward given the flexibility of our space. Our space uses a CFG, as in the excellent work of [1], but is distinct for several key reasons:

- Our space unifies multiple architectural families (ConvNets, Transformers, MLP-only) into **one single expressive space** while [1] present variations of their spaces centred around ConvNets only, with a separate instantiation focusing only on Transformers.
- einspace extends to probabilistic CFGs. This constitutes a significant contribution by enabling a set of benefits that include (i) allowing experts to define priors on the search space via probabilities, and (ii) enabling a broader range of search algorithms that incorporate uncertainty estimates inside the search itself.
- einspace contains recursive production rules (e.g. M &#8594; MM), meaning the same rule can be expanded over and over again, providing a very high flexibility in macro structures. [1] instead focuses on fixed hierarchical levels that limits the macro structure to a predefined (though very large) set of choices. We will ensure that these differences are better highlighted in the manuscript.

Further to this, we present a direct comparison to hNASBench201, the hierarchical CFG-based search space from [1]. These results show how einspace compares favourably to a different search space under the same evolutionary search. Overall, we highlight that our search results on einspace are competitive, even with far weaker priors on the search space.

Table A: Comparing einspace to hNASBench201 (from Schrodi et al. [1])
|| RE (hNASBench201) | RE (Mix) (einspace) |
|-|-|-|
| AddNIST | 93.82 | 97.72 |
| Language | 92.43 | 97.92 |
| MultNIST | 93.44 | 92.25 |
| CIFARTile | 58.31 | 62.76 |


**Experimental results**: We appreciate reviewer concerns regarding experimental evaluation, as this is a common problem in NAS work. To alleviate this concern we present an array of additional experimental results in our rebuttal, towards strengthening our submission. Our updated experimental evaluation now covers 16 different datasets, with sizes ranging from thousands of data points to over a million ('Satellite', NB360), and spatial resolutions of up to 256x256 (‘Cosmic’, NB360). Our updated experimental work provides further evidence of the efficacy of our proposed space, including more expensive tasks, and the experimental breadth can be considered more diverse than most previous NAS work.

Table B: Additional datasets from NAS-Bench-360 (one-dimensional)
| | WRN | DARTS (GAEA) | Expert | RE(WRN) einspace |
|-|-|-|-|-|
| Satellite | 15.29 | 12.51 | 19.80 | 12.55 |
| DeepSea | 0.45 | 0.36 | 0.30 | 0.36 |

[1] Construction of Hierarchical Neural Architecture Search Spaces based on Context-free Grammars, Schrodi et al, NeurIPS 2023.

---

### Comment · Area_Chair_PWx6 · 2024-08-09
**Read the rebuttal and discuss with the authors**

Hi Reviewers,

The authors have submitted their rebuttal responses addressing the concerns and feedback you provided. We kindly request that you review their responses and assess whether the issues you raised have been satisfactorily addressed.

If there are any areas where you believe additional clarification is needed, please do not hesitate to engage in further discussion with the authors. Your insights are invaluable to ensuring a thorough and constructive review process.

Best
AC

---

### Decision · Program_Chairs · 2024-09-25

**Decision:**

Accept (poster)

**Comment:**

The paper received reviews from six reviewers: three weak accepts, one borderline accept, and two borderline rejects. Most reviewers recognized the significance of expanding the NAS search space, which is a valuable contribution to the NAS field. The main concerns raised were related to the searching strategy and the need for more comprehensive evaluation on larger datasets.

The authors performed exceptionally well in their rebuttal, addressing most concerns by presenting additional evaluation results and explanations. Although two reviewers initially shown strong rejection attitude, they increased their scores after considering the rebuttal.

After reviewing the paper and the rebuttal, AC believes that the overall strengths outweigh the weaknesses and recommends acceptance.  This decision has been discussed with SAC. For the camera-ready version, the authors should incorporate all key results presented in the rebuttal and make significant improvements to the paper’s presentation.